# A redox-based electrogenetic CRISPR system to connect with and control biological information networks

Narendranath Bhokisham[1,2,5], Eric VanArsdale[2,3,4,5], Kristina T. Stephens[2,3,4], Pricila Hauk [2], Gregory F. Payne[2,3,4] & William E. Bentley [2,3,4 ✉]

Electronic information can be transmitted to cells directly from microelectronics via electrode-activated redox mediators. These transmissions are decoded by redox-responsive promoters which enable user-specified control over biological function. Here, we build on this redox communication modality by establishing an electronic eCRISPR conduit of information exchange. This system acts as a biological signal processor, amplifying signal reception and filtering biological noise. We electronically amplify bacterial quorum sensing (QS) signaling by activating LasI, the autoinducer-1 synthase. Similarly, we filter out unintended noise by inhibiting the native SoxRS-mediated oxidative stress response regulon. We then construct an eCRISPR based redox conduit in both *E. coli* and *Salmonella enterica*. Finally, we display eCRISPR based information processing that allows transmission of spatiotemporal redox commands which are then decoded by gelatin-encapsulated *E. coli*. We anticipate that redox communication channels will enable biohybrid microelectronic devices that could transform our abilities to electronically interpret and control biological function.

[1] Biological Sciences Graduate Program—College of Computer, Mathematical and Natural Sciences, University of Maryland, 4066 Campus Drive, College Park, MD 20742, USA. [2] Institute of Bioscience and Biotechnology Research, University of Maryland, 5115 Plant Sciences Building, College Park, MD 20742, USA. [3] Fischell Department of Bioengineering, A. James Clark Hall, University of Maryland, College Park, MD 20742, USA. [4] Robert E. Fischell Institute for Biomedical Devices, University of Maryland, Room 5102, A. James Clark Hall, College Park, MD 20742, USA. [5] These authors contributed equally: Narendranath Bhokisham, Eric VanArsdale. ✉email: bentley@umd.edu

The goal of communication is the efficient transmission of information. This is the case not just in electronics but also in biology where electromagnetic transmission is replaced by the flow of ions and molecules. Both electromagnetic and biomolecular communication channels must overcome data structure and noise limitations. Information content that is embedded into biomolecular structure is analogous to the data compression tools that we commonly employ in electronics. Biological error correction (including evolution), redundancy, and parallel processing align with techniques for sustaining data transmission in noisy environments. Noting that redox modalities in biology embrace electron movement, molecular structure, and reactivity, we suggest that redox also enables direct transfer of information (in the form of electrons) between biological and electronic systems. Moreover, this modality conforms nicely with the theoretical underpinnings of information transfer[1,2] because programmable electrical inputs directly mediate chemical transmissions that, in turn, are received and interpreted by biomolecules and cells.

Establishing a facile electronic to molecular communication channel that functions in biological systems would be transformative. For decades, electrical impulses have been applied to modulate biological function. Electromagnetic neural and muscular stimulation[3], alleviation of pain[4], wound healing[5] are some of the examples of bioelectronics that have already benefitted human health. However, in these instances, applied electrical impulses modulate ion-based currents[6] leading to systems level changes. Recently, an alternative strategy has been proposed wherein electrode-imposed information is transmitted locally, at the cellular level through an electron-based redox modality that embraces the structural features of molecular communication[7,8]. Redox is one of the most prevalent naturally occurring modes of communication in biology with implications in the gut microbiome[7,9], inflammation and autoimmunity[10,11], aging[12], and bacterial quorum sensing[13,14]. Redox is enabled by an array of small molecule redox mediators, such as ascorbate, NAD(P)H, hydrogen peroxide, superoxide, hydroxyl radicals and, more importantly, these redox mediators can be activated by the acceptance or donation of electrons. By this virtue, these molecules can transmit electronic information from electrodes to biology and vice versa[15–18]. In addition, nature has evolved a vast set of redox responsive regulatory elements, such as SoxS[19], OxyS[20], NFκB[21], Nrf2/Keap1[22], and these can be repurposed to respond to user-imposed redox-inputs. We have shown rewiring of two such regulators, SoxS[23] and OxyS[24–27], to enable user-imposed redox inputs for actuating and controlling native bacterial functions, such as motility and cell–cell signaling.

In this work, we use a CRISPR-mediated tool set to bring the abundance of CRISPR functions into redox communication channels. We assemble a two-part genetic system consisting of (i) SoxS-based electrogenetic promoters that are activated by redox-based molecular signaling; and (ii) CRISPR-Cas9 based synthetic transcriptional factors for multiplexed activation and inhibition. We refer to this as an electrogenetic CRISPR system (eCRISPR, Fig. 1) where the facile and robust programmability of electronics gains access to several functions within biology. First, oxidized pyocyanin (PYO), a phenazine class antibiotic produced by *Pseudomonas aeruginosa* is used to activate SoxS promoters. Usually, under aerobic conditions, PYO oxidizes the SoxR repressor leading to expression from SoxS promoters[13,23] and the reduced PYO, in turn, is likely re-oxidized by intracellular electron transfer. Under anaerobic conditions, PYO recycling is mediated by redox active electron acceptors such as ferricyanide (Fcn(O)) whose redox state is controlled by an external electrode[23] leading to control over gene expression in a permissive host having an attenuated oxidative stress response. The second

part consisted of CRISPR-Cas9 derived transcriptional factors that enable precise genome targeting and editing[28,29] and those containing a dead Cas9 (dCas9) enable precise gene silencing (referred to as CRISPRi)[30,31]. When dCas9 is fused with transcriptional activators such as VP16[32], the ω subunit of bacterial RNA polymerase[33], and SoxS[34], CRISPR has been shown to activate transcription (referred to as CRISPRa).

We hypothesize that by coupling the CRISPR-based synthetic transcriptional regulators with electrogenetic[23,35] promoter systems one would enable direct electrical control over host genome transcription. Moreover, because electronic control is mediated by simply biasing electrodes that are proximal to cell-containing media, the vast repertoire of electronic signaling and control is made accessible. Tunability could be achieved by controlling size, location, and electrode material[36–39], programmable chemical gradients could be created (e.g., electrophoresis, gelation)[40,41] and complex signal inputs can be applied to employ advanced signal processing methodologies on biological circuits[17]. To this end, we work on the previously described bacterial CRISPRa system by Bikard et. al. involving the use of dCas9-ω as transcriptional activator[33] and built an electrically tunable and controllable CRISPRa system (eCRISPR). We tune various factors that govern the stoichiometries of the CRISPR components and later integrated them with electrogenetic SoxS promoters. Upon electrical induction, we observe a ~15-fold increase in transcriptional activation of fluorescent proteins. To demonstrate controlled biological function, we use eCRISPR to amplify cell-to-cell communication among bacteria. Through these demonstrations, we show electronically programmed biological communication between bacterial cells.

In addition to building an *E. coli*-specific electrogenetic controller, we sought to demonstrate function in other non-compliant hosts[23,42]. That is, by showing that eCRISPR can work with gRNA selected for different hosts, we demonstrate an information channel that enables communication of different messages to multiple participants through the same electronic input. Our previous work[23] exploited DJ901 *E. coli* that have an attenuated stress response; this enabled more focused amplification of SoxS-mediated gene transcription. In effect, suppression of the native stress response acts as a filter, reducing noise[43,44] and streamlining transmission. We and others have previously demonstrated that by downregulating concomitant pleotropic responses, one can focus metabolic activity towards the desired genetically engineered functions, including targeted gene expression[45–50]. The SoxS promoter used in this study is part of the SoxRS regulon, a global regulator of the oxidative stress defense response in *E. coli*. Upon exposure to oxidative stressors, a several-fold upregulation of SoxS[51] leads to upregulation of ~15 genes including superoxide dismutases (*sodA* and *sodB*), fumarate hydratase (*fumC*) among others, endowing cells with broad and effective means for alleviating oxidative stress. Here, we use CRISPR-Cas9 systems to selectively and transiently silence genomic SoxS and the ensuing oxidative defense responses; this reduction in pleiotropic noise enabled a 4-fold increase in targeted electrogenetic SoxS promoter activity in *E. coli*. Also, portability of the electrogenetic system is shown by performing analogous experiments in *Salmonella enterica*, where we found a similar 4-fold increase in expression from *E. coli* based *soxS* promoters, but with repression targets designed for *Salmonella*.

Finally, we show eCRISPR as both a signal amplifier and filter that results in better congruency between electrical input and cellular output in complex environments. To do this, we assemble a bio-electronic interface and demonstrate (i) the electrochemical formation of biochemical gradients, and (ii) how these gradients spatially guide eCRISPR-mediated function within cells immobilized in hydrogels. CRISPR-containing cell populations

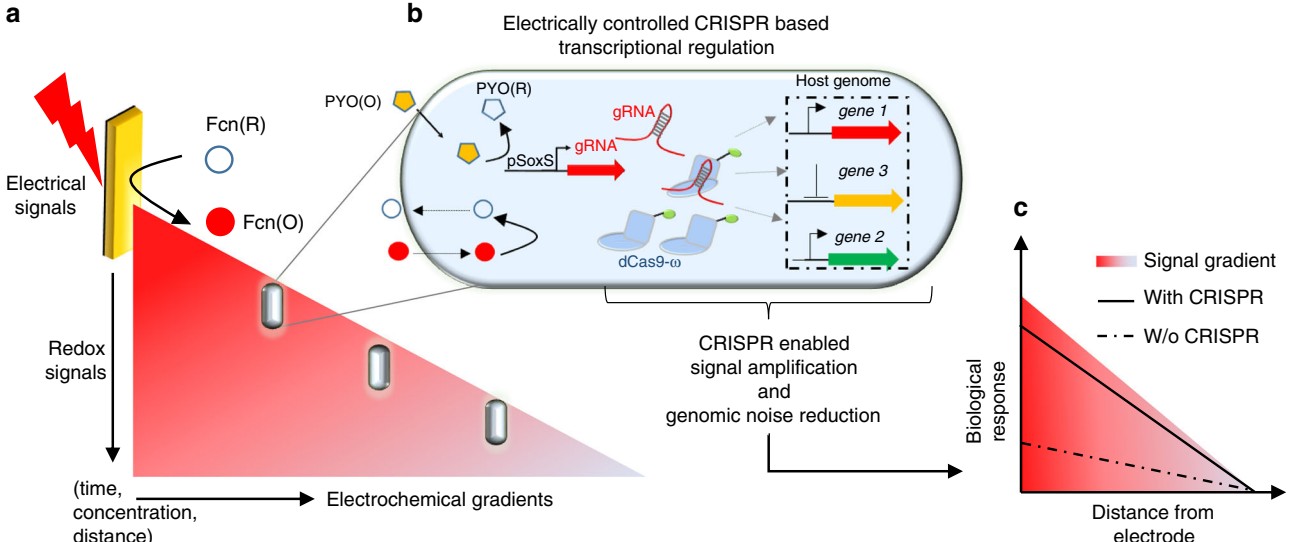

**Fig. 1 Scheme of the eCRISPR system. a** A scheme of the spatio-temporal electrochemical gradients generated at the bio-electronic interface where an oxidizing potential is applied via an external electrode to oxidize the Fcn resulting in generation of a Fcn(O) gradient across space and time away from the electrode. **b** The eCRISPR system comprising of the SoxS-based electrogenetic promoter is activated by pyocyanin (PYO) and the redox mediator ferricyanide Fcn(O), resulting in expression of gRNAs and the formation of CRISPR-based transcriptional factor complexes. These complexes can be directed to desired sites leading to properties of signal amplification and noise reduction in complex signal gradients at the bio-electronic interface. **c** Resulting effect of the eCRISPR based genomic regulation is that biological populations display more linear and amplified biological responses in response to signal gradients leading to higher signal fidelity across the bio-electronic interface.

displayed enhanced signal strength relative to cells without eCRISPR control. In this way, we demonstrate the flexibility of electronic control over cell function and the benefits provided by eCRISPR genetic tuning.

## Results

**A tunable CRISPRa system in bacteria**. We first intended to create an inducible and tunable CRISPR-Cas9 mediated transcriptional activation (CRISPRa) system that enabled integration with bacterial quorum sensing (QS) signal transduction systems based on *E. coli* W3110[52–58]. To do this, we created NB101 strain (*E. coli* W3110 Δ*rpoZ, lacZ*, Supplementary Table 1) for adapting a previously reported bacterial CRISPRa system wherein the RNA polymerase subunit ω (*rpoZ*) was genetically fused to *S. pyogenes* origin deactivated Cas9 (dCas9) resulting in 23-fold transcriptional activation of green fluorescent protein (GFP)[33]. Importantly, in NB101, we found comparable levels of CRISPRa (Supplementary Fig. 1) and used this strain for further experiments.

We next focused on improving the levels of transcriptional activation. Bikard *et.al.* had used native *S. pyogenes* promoters to express CRISPR components and employed tracrRNA:crRNA hybrids to present the spacers for CRISPRa of GFP[33]. Instead of tracrRNA:crRNA hybrids, we introduced spacers (W108 spacer[33]) in the form of single short gRNA (denoted 108 gRNA) expressed from a strong constitutive promoter (J23119) in p108gRNA. Additionally, we placed the dCas9-ω fusion under an inducible Tet promoter in pdCas9-ω and used the leaky expression of dCas9-ω to enable GFP activation from pWJ89. This combination resulted in ~5-fold greater GFP activation, suggesting that the relative levels of dCas9-ω and gRNA played an important role in controlling transcription (Supplementary Fig. 2). To further enhance CRISPRa, a common strategy used in eukaryotic CRISPRa systems is to increase transcriptional activator domains per dCas9 molecule[32], however increasing the number of ω did not result in enhanced CRISPRa (Supplementary Fig. 3) in

agreement with a recent study[59]. To enable a tunable bacterial CRISPRa system, we first employed mismatches in the first few base pairs of the gRNA that first anneals to the DNA region adjacent to the protospacer adjacent motif (PAM), also referred to as the seed region of spacer[60]. We observed no role for mismatches in CRISPRa tunability (Supplementary Fig. 4).

We then focused on using inducible promoters to tune expression of dCas9 and gRNA. First, we induced the expression of dCas9-ω from Tet promoters by addition of anhydrotetracycline (aTc) (Supplementary Fig. 5). While dCas9-ω increased with aTc, CRISPRa and GFP fluorescence did not. Additional attempts to reduce leaky expression by modulating dCas9-ω from the Tet promoter[61] were unsuccessful (Supplementary Fig. 6); henceforth we relied on leaky expression of dCas9-ω. We next focused on gRNA by moving from the constitutive J23119 promoter to inducible promoters (pTrc in pTrc-108gRNA and pSoxS in pSoxS-108gRNA) and while the gRNA expression increased with increased inducer concentration (Supplementary Fig. 7b), there was no change in CRISPRa (Supplementary Fig. 7c, d). These results suggest there was sufficient gRNA expressed from both promoters, the net result being a saturated CRISPRa response.

We next replaced the high copy pBR322 origin in the gRNA expressing pSoxS plasmid with a low copy pSC101* origin to create plasmid pSC-S108gRNA. Conversely, we replaced the pSC101* origin of the target GFP plasmid, pWJ89, with the pBR322 origin to create pMC-GFP (Fig. 2a). With this rearrangement, we induced gRNA expression from the SoxS promoter. We observed increased gRNA expression after 6 h (Fig. 2b) with increasing PYO, as well as a ~21-fold increase in GFP fluorescence (Fig. 2c), indicating the tunability of CRISPRa.

Having shown that we could tune CRISPRa by controlling the gRNA levels and the number of gene targets, we next sought to see whether CRISPRa system could be used in activation of QS based cell–cell communication. (Fig. 2a, d). We used the *E. coli* CRISPRa system to activate transcription of LasI, a QS autoinducer-1 synthase from *Pseudomonas aeruginosa*. To do this, we replaced the *gfpmut2* in pMC-GFP with the *lasI*

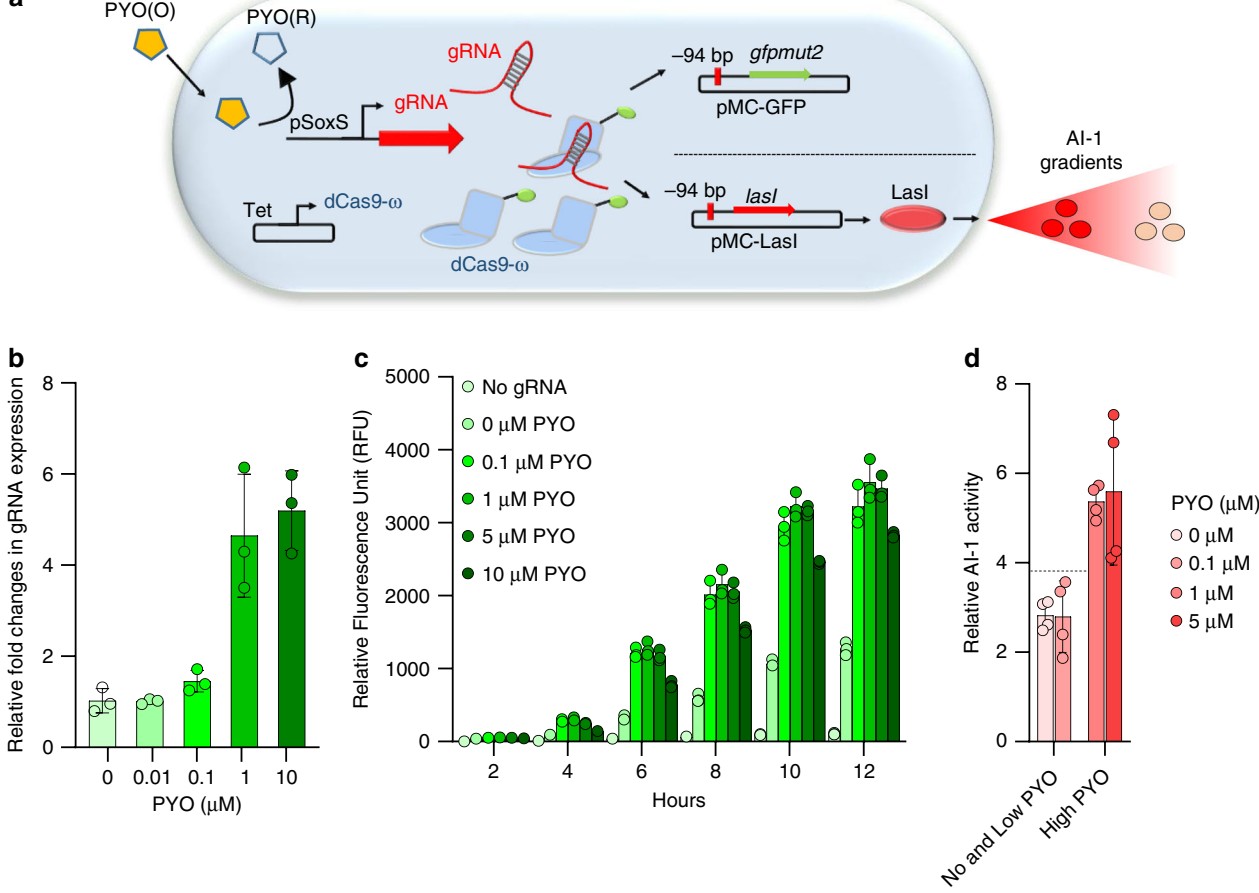

**Fig. 2 Tunable CRISPRa system by controlling gRNA expression from SoxS promoters. a** Scheme of a tunable CRISPRa system with the *E. coli* pSoxS promoter expressing gRNAs for transcriptional activation. Transcriptional activator dCas9-ω is expressed under a Tet promoter in pdCas9-ω which then combined with gRNAs from pSC-S108 to target *gfpmut2* and *lasI* resulting in transcriptional activation of GFP and LasI in NB101 cells. Expression of LasI results in synthesis of autoinducer-1 (AI-1). **b** qPCR data indicating the relative levels of gRNAs after 6 h of promoter induction. **c** CRISPRa mediated GFP fluorescence across 12 h measured via a plate reader. **d** AI-1 assay indicating the amounts of AI-1 generated via CRISPRa. AI-1 producer cells are induced with different PYO concentrations and after 4 h of induction, conditioned media from the AI-1 producer cells is collected and added to the AI-1 reporter cells. After 4 h of incubation with reporter cells, bioluminescence is measured via a luminometer. In all figs, independent experimental replicates are represented by circles and height of the bars indicate mean. For **b** and **c**, $n = 3$ and for **d**, $n = 4$. Error bars indicate standard deviation. Source data for all figures is provided separately.

(pMC-LasI) and transformed it along with pSC-S108gRNA and pdCas9-ω into NB101. We referred to these populations as AI-1 producer cells. We added different levels of PYO at the time of re-inoculation into LB media under aerobic conditions and activated CRISPRa of LasI. After 4 hours, we collected conditioned media (CM) and incubated with AI-1 reporter cells (see "Methods"). Results (Fig. 2d) indicated that AI-1 activity, a measure of QS activity, was increased with PYO. Notably, the higher levels of PYO increased the QS activity the greatest.

Thus, in Fig. 2, we showed that by replacing dCas9-ω expression from native *S. pyogenes* promoters to enable leaky expression from Tet promoters and by switching from the tracrRNA: crRNA hybrid system to a short gRNA system using synthetic promoters, we improved the CRISPRa response by ~5-fold. Tunability in this CRISPR system was then achieved by varying the stoichiometric ratios of not just gRNA but also the number of targets in the system resulting in ~21-fold increase in transcription of fluorescent reporters. We further demonstrated CRISPRa-mediated QS communication across two different populations by actuating AI-1 synthesis. We found a ~7-fold increase in bioluminescence among AI-1 responding populations.

**Electrically tunable CRISPRa—eCRISPR.** Having demonstrated small molecule-based induction of a tunable CRISPRa system, we next sought to electrically induce CRISPRa using the SoxS electrogenetic promoter[23], creating the eCRISPR system (Fig. 3a). We started by testing anaerobically with redox mediators, PYO and Fcn. We replaced the reporter gene from *gfpmut2* in pMC-GFP with *phiLOV* (pMC-phiLOV) that is capable of fluorescing under both aerobic and anaerobic conditions[62] and used it as a target for CRISPRa (Fig. 3a). As shown above (Fig. 2), under aerobic conditions, PYO alone is sufficient to oxidize the SoxR repressor and activate gene expression from SoxS. However, under anaerobic conditions, PYO recycling of the SoxR requires the use of redox active electron acceptors such as Fcn(O). Hence going forward, under anaerobic conditions, both PYO and Fcn were added to mediate activation of the SoxS promoter. We grew NB101 cells harboring plasmids with the tunable CRISPRa plasmids (pSC-S108gRNA, pdCas9-ω) and the new reporter plasmid pMC-phiLOV in LB media at 37 °C under aerobic conditions. At $OD_{600}$ 0.6, we washed the cells and resuspended them in minimal-M9 media and performed further experiments under anaerobic conditions. To characterize gRNA expression from SoxS promoters and the resulting CRISPRa, we first optimized the concentrations

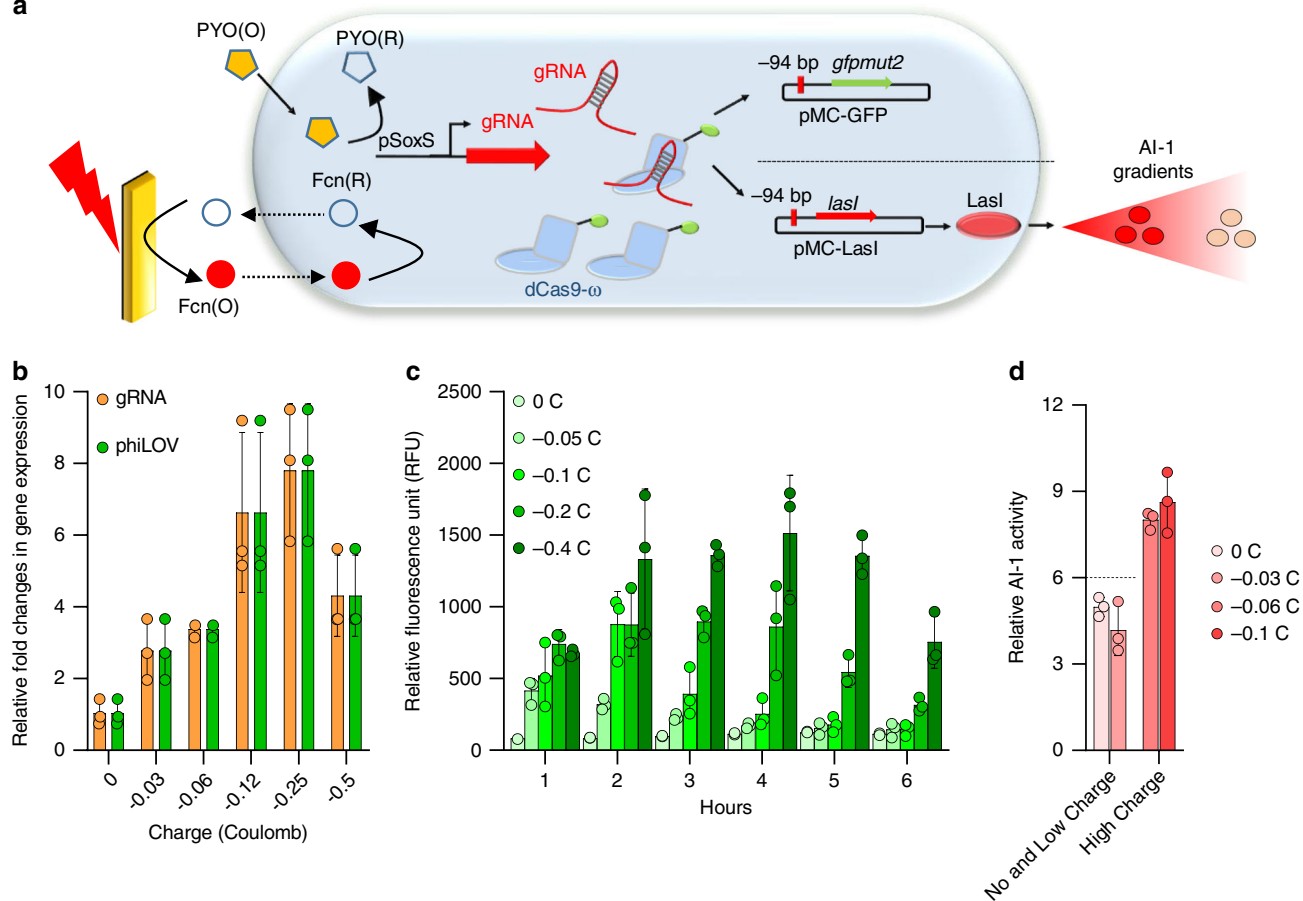

**Fig. 3 Electrically tunable CRISPRa—eCRISPR. a** Scheme of an electrically controlled CRISPRa system in bacteria with electrical control over gRNA expression from the *E. coli* pSoxS promoter. NB101 cells containing plasmids pSC-S108gRNA, either pMC-phiLOV or pMC-LasI and pdCas9-ω are grown in aerobic conditions to $OD_{600}$ 0.6 and later resuspended in minimal-M9 media and moved to an anaerobic chamber for electrical induction. PYO and Fcn(R) is added to cells and an oxidation potential of +0.5 V is applied to oxidize Fcn(R) to Fcn(O) and turn ON the pSoxS promoter. **b** qPCR data indicating the relative gRNA and *phiLOV* expression levels after 2 hours of electrical induction. **c** CRISPRa mediated phiLOV fluorescence with varying levels of electrical induction measured via flow cytometry. **d** AI-1 assay indicating the amounts of AI-1 generated via eCRISPRa. AI-1 producer cells are induced with different amounts of electric charges and after 2 h of induction, conditioned media from the AI-1 producer cells is collected and added to the AI-1 reporter cells. After 4 h of incubation with reporter cells, bioluminescence is measured via a luminometer. In all figures, independent experimental replicates ($n = 3$) are represented by circles. Height of the bars indicate mean and error bars indicate standard deviation. Source data for all figures is provided separately.

of redox mediator Fcn(O) and PYO. phiLOV fluorescence increased with increasing concentrations of Fcn(O) and PYO (Supplementary Fig. 8a, b). We found the highest PYO and Fcn(O) concentrations (10 µM and 50 mM respectively) yielded ~18-fold CRISPRa (Supplementary Fig. 8c). However, to maintain linearity in fluorescence measurements, we subsequently used 5 µM PYO and 50 mM Fcn for further experiments.

Next, we sought to demonstrate electrode-mediated control of CRISPRa. Instead of adding oxidized Fcn(O) to cells, we added 5 µM of PYO and 50 mM of Fcn(R) and applied an oxidizing potential of +0.5 V using the electrochemical setup ("Methods") to oxidize Fcn(R) to Fcn(O). We applied varying amounts of electric charge to cell populations and transferred them to a 37 °C incubator for 2 h. We then measured the relative expression levels of gRNA from the SoxS promoter and phiLOV from CRISPRa (Fig. 3b). With increased applied charge, we found increased levels of both the gRNA as well as phiLOV, until −0.5 Coulomb when both decreased perhaps due to expression-mediated toxicity. Figure 3c indicates the fluorescence obtained during 6 h after electrical induction with varying levels of applied charge. As anticipated, fluorescence increased with charge for the first 4 h

up to −0.4 C, with a maximum of ~13-fold transcriptional activation.

As a demonstration of information transfer between electronic to biological signaling modalities, we attempted to electronically elevate QS communication using the AI-1 producer cells that transduce electric signals into biologic signals. We used the CRISPRa system to activate transcription of LasI (see Fig. 3a). At $OD_{600}$ of 0.6, we moved the AI-1 producer population to minimal-M9 media and under anaerobic conditions applied varying amounts of electrical charge. After 2 h of incubation at 37 °C, we collected the conditioned media (CM) and incubated this CM with AI-1 reporter cells for 4 h under aerobic conditions and measured relative QS activity via bioluminescence (Fig. 3d). Results indicated there was a ~1.5-fold increase in QS activity with an applied electric charge greater than −0.06 C. That is, cells already communicating via AI-1 based communication (an appreciable background level of AI-1 in the zero due to leaky expression and low charge cases) received a significant boost in their signal by the contributions from the electrogenetically stimulated producer cells when the charge exceeded −0.06 C. Thus, these results clearly demonstrate electronic actuation of

eCRISPR to mediate native signaling between communicating bacterial populations. We note further, that in our previous work we used a genetically modified host in which the SoxS-regulated stress response regulon was attenuated. We subsequently hypothesized that to enhance this process further we might suppress pleiotropic host responses and thereby focus metabolic activity on the electrogenetic target.

**Multiplexing to attenuate noise and demonstrate portability.** To increase the signal strength in the bio-electronic communication mediated by SoxS-based redox promoters, we sought to utilize CRISPR to suppress noise caused by metabolically prohibitive oxidative stress responses otherwise triggered by electrical signals, thus leading to amplification of the desired signals. Also, to mediate widespread applicability and to generalize electronic to biological information transfer, we examined portability of the electrogenetic promoter systems from *E. coli* and *Salmonella* with minimal rewiring of genetic circuits (Fig. 4a).

First, we focused on eCRISPR as a reliable signal amplifier. Pyocyanin (PYO), the inducer of SoxS promoter is also an oxidative stressor[23]; *E. coli* attenuates oxidative stress primarily by two global transcriptional regulators, SoxS and OxyS[63]. Here we sought to determine whether CRISPR-mediated repression of SoxS in the *E. coli* genome could lead to an improvement in SoxS promoter activity of plasmid-encoded transgenes and, in so doing, act as a electrogenetic signal amplifier. We expected that the repression of SoxS in the genome would have no effect on the pSoxS promoters present in the various CRISPRa constructs because SoxR drives the control of SoxS promoter and SoxR is untouched in our experiments.

We repurposed the transcriptional activator dCas9-ω for repression of SoxS-mediated oxidative stress responses by targeting *soxS* in the bacterial genome. The ideal dCas9 target site for repression is the −35 to −10 promoter region[33], however, since we used the *E. coli soxS* promoter for electrical activation in the tunable gRNA plasmid pSC-S108, we chose to target a PAM site that was closest downstream site to the *soxS* transcriptional start site (TSS) in the genome. We targeted two PAM sites in the non-coding strand of the *soxS* gene at +4 bp and +5 bp and designed two corresponding gRNAs S-1 and S-2 ("Supplementary Methods"). We expressed these gRNAs, with a nonspecific control, under a strong constitutive promoter, J23119, and provided dCas9-ω via leaky expression from the Tet promoter as noted above. As a positive control for CRISPRi, we used dCas9 devoid of the ω subunit, a known transcriptional repressor[33] using pdCas9. We transformed the respective gRNA and dCas9 plasmids into NB101 cells grown in LB media at 37 °C. At OD600 0.6, we induced *soxS* in the genome with 5 μM PYO in aerobic conditions and after 3 h, collected RNA and performed qPCR. We compared the relative levels of *soxS* expression under the presence of S1, S2 and control gRNAs in combination with both dCas9 and dCas9-ω (Fig. 4b). With control gRNAs, there was a ~12 to 17-fold increase in *soxS* expression compared to the no PYO condition. Importantly, with S1 and S2 gRNA, there was no increase in *soxS* upon addition of PYO, demonstrating effective CRISPRi. Since *soxS* repression from S1 gRNA was marginally better than S2, we used S1 in all further experiments. Also, since there was no difference in repression levels between dCas9 and dCas9-ω, we proceeded with dCas9-ω for further experiments.

Our objective was the repression of *soxS* in the genome via CRISPR and thereby obviating the upregulation of various SoxS-regulated defense response genes, a potential metabolic burden for the cells in future electrogenetic applications. Using qPCR, we measured the expression levels of two genes highly upregulated by SoxS: *sodA* (superoxide dismutase A) and *fumC* (fumarate

hydratase). Under similar conditions as in Fig. 4b, S1 and dCas9-ω prevented upregulation of both *sodA* and *fumC* (Supplementary Fig. 9). To confirm, we added 0.5 mM paraquat, a strong and well-characterized oxidant[19]. We found ~75-fold activation of the *soxS* with the control gRNAs (Supplementary Fig. 10a), but in the presence of *soxS* specific S1 gRNA and dCas9-ω, *soxS* was repressed to background levels. Similarly, *sodA* (Supplementary Fig. 10b) and *fumC* (Supplementary Fig. 10c) were not upregulated due to the absence of elevated SoxS.

Next, we studied if the repression of *soxS* in the *E. coli* genome lead to improvements in plasmid-encoded SoxS promoter activity. We used a previously described electrogenetic reporter plasmid pTT01[23] origin containing *phiLOV* under the SoxS promoter with a pBR322 origin (denoted pSoxS-phiLOV) and inserted a cassette for the S1 gRNA sequence to be expressed under the J23119 constitutive promoter (creating pSoxS-phiLOV: pS1gRNA). Under anaerobic conditions, we induced the NB101 cells harboring pSoxS-phiLOV:pS1gRNA with 5 mM Fcn(O) and 5 μM PYO and measured phiLOV fluorescence levels via flow cytometry. As controls, we also transformed just pSoxS-phiLOV into DJ901 cells (ΔsoxRS). In Fig. 4c, phiLOV fluorescence measurements indicated that the addition S1 gRNA and dCas9-ω along with the pSoxS-phiLOV reporter plasmid led to a 3 to 4-fold increase in NB101 cells (WT for *soxS*). This was comparable to genetically modified DJ901 (ΔsoxRS) cells (Fig. 4c). We also engineered expression of S1 gRNA under a SoxS promoter with a pSC101* origin in plasmid pSCSoxS-S1gRNA. Here too, we observed a similar ~3-fold increase in phiLOV fluorescence. These results demonstrate that upregulation of the plasmid-encoded SoxS promoter was indeed enhanced by the concomitant downregulation of genomic SoxS.

Next, we focused on the use of CRISPR for constructing a portable electrogenetic signaling system by transferring these components to *Salmonella*. We designed specific gRNAs to repress *soxS* in *S. enterica* serovar Typhimurium LT2 and demonstrated enhanced electrogenetic promoter output from the *E. coli* SoxS promoters. First, we identified a PAM site downstream of the TSS at +29 bp in the non-coding strand of the *soxS* gene, denoted S3 gRNA (sequence in "Supplementary Methods"). We replaced the gRNA sequence in plasmid pSoxS-phiLOV:pS1gRNA to create pSoxS-phiLOV:pS3gRNA and transformed it along with pdCas9-ω into *Salmonella* LT2. We grew the cells to OD600 0.6 under aerobic conditions and induced the *soxS* in the genome with 5 μM PYO. After 3 hours, we collected RNA samples and performed qPCR. As controls, we also had *Salmonella* without the S3 gRNA and dCas9-ω. Upon addition of PYO, in controls there was a ~14-fold increase in *soxS* gene expression. However, with S3 gRNA and dCas9-ω, there was no significant increase in *soxS* indicating successful CRISPR-based repression of *soxS* in the *Salmonella* genome (Supplementary Fig. 11). Next to verify whether the repression of *soxS* in the genome would lead to an increase in *E. coli* SoxS promoter activity, we grew *Salmonella* LT2 cells containing pSoxS-phiLOV: pS3gRNA and pdCas9-ω to OD600 0.6 in LB media under aerobic conditions and later resuspended the cells in minimal-M9 media and moved them to anaerobic conditions. We then added 10 μM PYO and 5 mM Fcn(R) and applied a +0.5 V oxidizing potential. After a discharge of −0.5 C, we moved the cells to 37 °C inside the anaerobic chamber and measured phiLOV fluorescence over the ensuing 6 h. As controls, we performed similar experiments without the gRNA. Results (Fig. 4d) revealed minimal expression overall without the electronic charge, while fluorescence reached 300–800 (a.u.) after an applied charge of −0.5 C. Notably, the pSoxS-phiLOV results demonstrated up to a 4-fold increase in gene expression without repression of *soxS*. In the cases with dCas9-ω and *soxS* guide RNA, the *soxS* driven gene expression

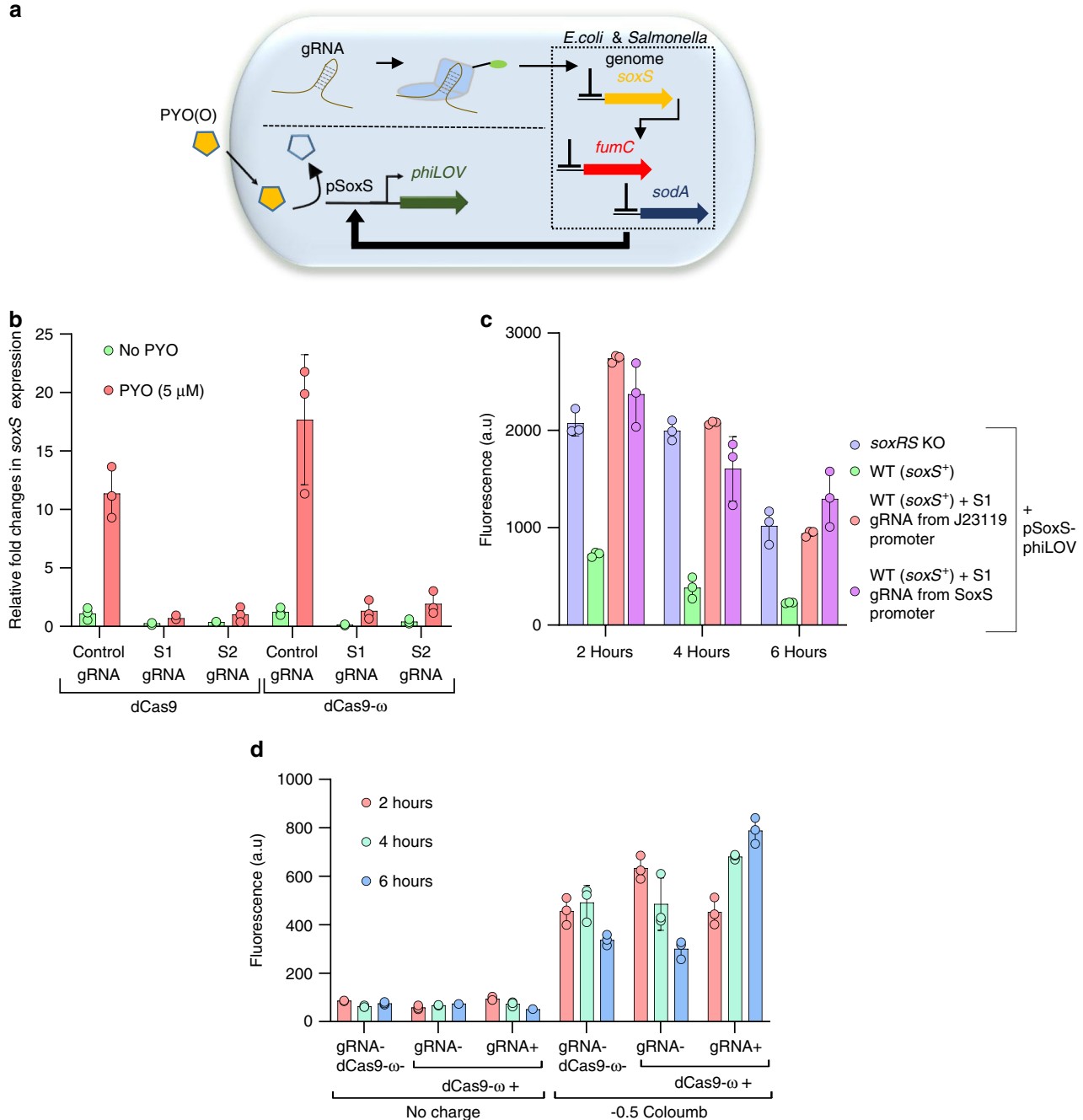

**Fig. 4 eCRISPR multiplexing to attenuate noise and demonstrate portability. a** Scheme of the CRISPR-mediated repression of *soxS* in the genome leading to an enhancement in the SoxS promoter activity. Two PAM sites in the *E. coli soxS* coding region are targeted with S1 and S2 gRNA's respectively. As controls, a non-specific control gRNA is used as well. All gRNA expressing plasmids were individually transformed into NB101 cells and both dCas9 and dCas9-ω are tested for gRNA mediated repression. **b** indicates relative fold changes in *soxS* expression after 3 h of induction with 5 μM PYO. **c** phiLOV fluorescence data indicating the increase in SoxS promoter activity when coupled with CRISPR-mediated *soxS* repression in the *E. coli* genome. Blue: DJ901 + pSoxS-phiLOV; Green: NB101 + pSoxS-phiLOV; Red: NB101 + pSoxS-phiLOV:pS1gRNA + pdCas9-ω; Violet: NB101 + pSoxS-phiLOV + pSCSoxS-S1gRNA + pdCas9-ω. Under anaerobic conditions, $OD_{600}$ 0.6 cells are induced for 6 h with 5 mM Fcn(O) and 5 μM PYO and phiLOV fluorescence is measured via flow cytometry. **d** Data indicating the CRISPR-mediated repression of *soxS* in the *Salmonella* genome leading to an enhancement in the *E. coli* SoxS promoter activity. S3 gRNA specific to the soxS coding region in *S. enterica* is engineered into pSoxS-phiLOV to create pSoxS-phiLOV:pS3gRNA. This plasmid is transformed along with pdCas9-ω into *Salmonella*. As controls, pSoxS-phiLOV without the gRNA and dCas9-ω is used as well. Cells are grown to $OD_{600}$ 0.6 in LB media, resuspended in minimal-M9 media and moved to an anaerobic chamber. For electrical induction, 5 mM Fcn(R) and 10 μM PYO is added to cells and an electric charge of −0.5 C is applied and phiLOV fluorescence is measured via flow cytometry for upto 6 h. In all figures, independent experimental replicates (*n* = 3) are represented by circles, height of the bars indicate mean and error bars indicate standard deviation. Source data for all figures is provided separately.

increased another 2–4-fold, especially in the 6 h samples. These results demonstrate that *soxS* repression in *Salmonella* genome resulted in an increased electrogenetic output from the *E. coli* *soxS* promoter in *Salmonella* as well. More importantly, these results indicated that our electrogenetic approach was portable to non-chassis strains such as *Salmonella* and that CRISPRi could be employed to target select genes in additional genomes leading to more diverse system designs.

Having shown that we could repurpose the dCas9-ω transcriptional activator to repress genes as well by simply targeting the dCas9-gRNA complexes downstream of a transcription start site, we sought to study whether CRISPRi and CRISPRa could be performed simultaneously at different sites using the same activator in *E. coli*. We expressed both 108 gRNA and the S1 gRNA as a single hybrid gRNA transcript from the same SoxS promoter. To mediate gRNA processing, we introduced self-cleaving ribozymes (see Supplementary Fig. 12a). We observed the phiLOV activating 108 gRNA to be functional resulting in phiLOV fluorescence (Supplementary Fig. 12b) and the *soxS* repressing gRNA to be functional (Supplementary Fig. 12c) resulting in an increase in phiLOV fluorescence from the SoxS promoter (Supplementary Fig. 12d). These results indicated (i) successful ribozyme mediated RNA processing resulting in gRNAs with different functionalities and (ii) that dCas9-ω can simultaneously perform transcriptional repression and activation at different sites. However, with multiplexed RNA's, while there was an increase in phiLOV fluorescence from the SoxS promoter, there was no increase in CRISPRa-mediated phiLOV fluorescence (Supplementary Fig. 12b) suggesting the limitations of this approach. More studies are needed to identify the factors that govern the linearity of both the SoxS-mediated enhancement of SoxS promoter activity as well as CRISPRa.

Importantly, results in Fig. 4 demonstrate that the transcriptional activator dCas9-ω can be repurposed for repression of oxidative stress defense response genes leading to a concomitant 3–4-fold overall enhancement in plasmid-encoded SoxS promoter activity in *E. coli* as well as *Salmonella*. From an information processing perspective, the oxidative stress response is analogous to noise and attenuation of this noise led to focused metabolic activity towards phiLOV expression; analogous to the amplification of information. The ease with which these systems can be transported across various species with minimal genome wiring indicates its portability.

**Spatiotemporal electronic control of eCRISPR**. To demonstrate the importance of eCRISPR as a signal amplifier and filter in spatiotemporal environments, we constructed a bio-device interface where electrical inputs are translated into a transient biochemical gradient. Then, biological signaling networks inside the cells respond to these external gradients depending on their location (Fig. 5a). We expected that the presence of CRISPR circuits would result in an overall biological signal amplification while still preserving the gradient nature of the informational signals that exist at the interface of electronic and biologic signaling networks.

First, we created an electrochemical gradient of Fcn. We filled the Owl Easycast B1A (Thermo Scientific) gel electrophoresis system with minimal-M9 media supplemented with 50 mM potassium ferrocyanide and 5 μM PYO. The apparatus was placed in an anaerobic chamber and a continuous +0.3 V potential was applied across the apparatus for 8 hours. The conversion of Fcn(R) to Fcn(O) was shown by measuring absorbance at 420 nm (Fig. 5b). A distinct ferricyanide gradient was revealed over time with the highest level of Fcn(O) being directly adjacent to the working electrode. As expected, these levels increased over time for the 8 h period.

Next, to study the effect of the ferricyanide gradients on electrogenetic cells, we immobilized cells within gelatin hydrogels, immersed these gels in media, applied charge for specified times, and evaluated their responses. Cells were grown in MOPS-M9 media to $OD_{600}$ 0.5, encapsulated into gelatin hydrogels and cast onto microscope slides (Methods). We placed the hydrogel laden slides (3 at a time, in parallel lanes) in the middle chamber of the gel apparatus (Fig. 5a) and applied a potential of +0.3 V for 8 h. The microscope slides were then removed and analyzed by confocal microscopy for phiLOV fluorescence (Supplementary Fig. 13). We tested various cell populations including W3110 cells with phiLOV reporter plasmids (pSoxS-phiLOV), NB101 cells with pdCas9-ω and phiLOV reporters, as well as NB101 cells with pdCas9-ω and pSoxS-phiLOV:pS1gRNA plasmids. Results indicated that in all cell populations, upon electrical induction for 8 h, cell fluorescence decreased with increasing distance away from the working electrode, corresponding to the ferricyanide gradient. Most notably, within the various cell types, NB101 cells with pdCas9-ω and pSoxS-phiLOV:pS1gRNA plasmids, carrying the SoxS specific gRNA had a nearly 2-fold increase in fluorescence at all distances. Also, the change in fluorescence for cells containing the CRISPRi mechanism decreased with distance (Fig. 5c). In Fig. 5d, we tested several conditions where either the PYO or the Fcn(O) or the electric charge was removed. As anticipated, there was no enhanced cell fluorescence after 8 h in all these cases. In some conditions, a mixture of 25 mM Fcn(O) and Fcn(R) each was added to avoid potential toxicity from high concentrations of Fcn(O) while still maintaining the total Fcn concentration at 50 mM as in the other control conditions. The only condition in which there was appreciable fluorescence was when 25 mM Fcn(O), 25 mM Fcn(R), and 5 μM PYO as well as the +0.3 V electric field were applied to the device. These data indicated that cells cast in 2% gelatin did not respond to spiked 25 mM Fcn(O) unless an external charge was applied over the 8 h duration (Fig. 5d). Both the conversion of Fcn(R) to Fcn(O) and Fcn(O) renewal were enabled in these tests by the presence of the electric field. This was reflected in the increase of fluorescence at the leading edge of this control in Fig. 5d.

Together, these data support the conclusion that eCRISPR provides an enhanced response to electrochemically created spatial gradients of ferricyanide and that these gradients can control cell function across large length scales and within immobilized cells. As a result, eCRISPR can be used to develop congruent spatiotemporal responses within a transmission gradient that non-engineered cells struggle to decode.

## Discussion

In this work, we first varied the stoichiometries of several CRISPRa components, leading to development of a tunable and inducible CRISPRa system. We next demonstrated electrogenetic control by transitioning actuation to the SoxRS regulon; a ~15-fold electrogenetic transcriptional activation was found. Third, we repurposed the dCas9 based transcriptional activator to repress the host's oxidative stress defense responses leading to 3–4-fold increase in SoxS promoter output in both *E. coli* and *Salmonella*. Finally, we showed using a gel electrophoresis system, that gene expression could be spatiotemporally controlled via simple electronics.

Also, while a tunable CRISPRa system was found by manipulating gRNA expression, efforts to control expression via manipulation of dCas9 were unsuccessful. In addition, the inherent leaky expression of dCas9 from Tet promoters was more than enough for transcriptional activation and further increases in dCas9 decreased target gene expression. This problem of

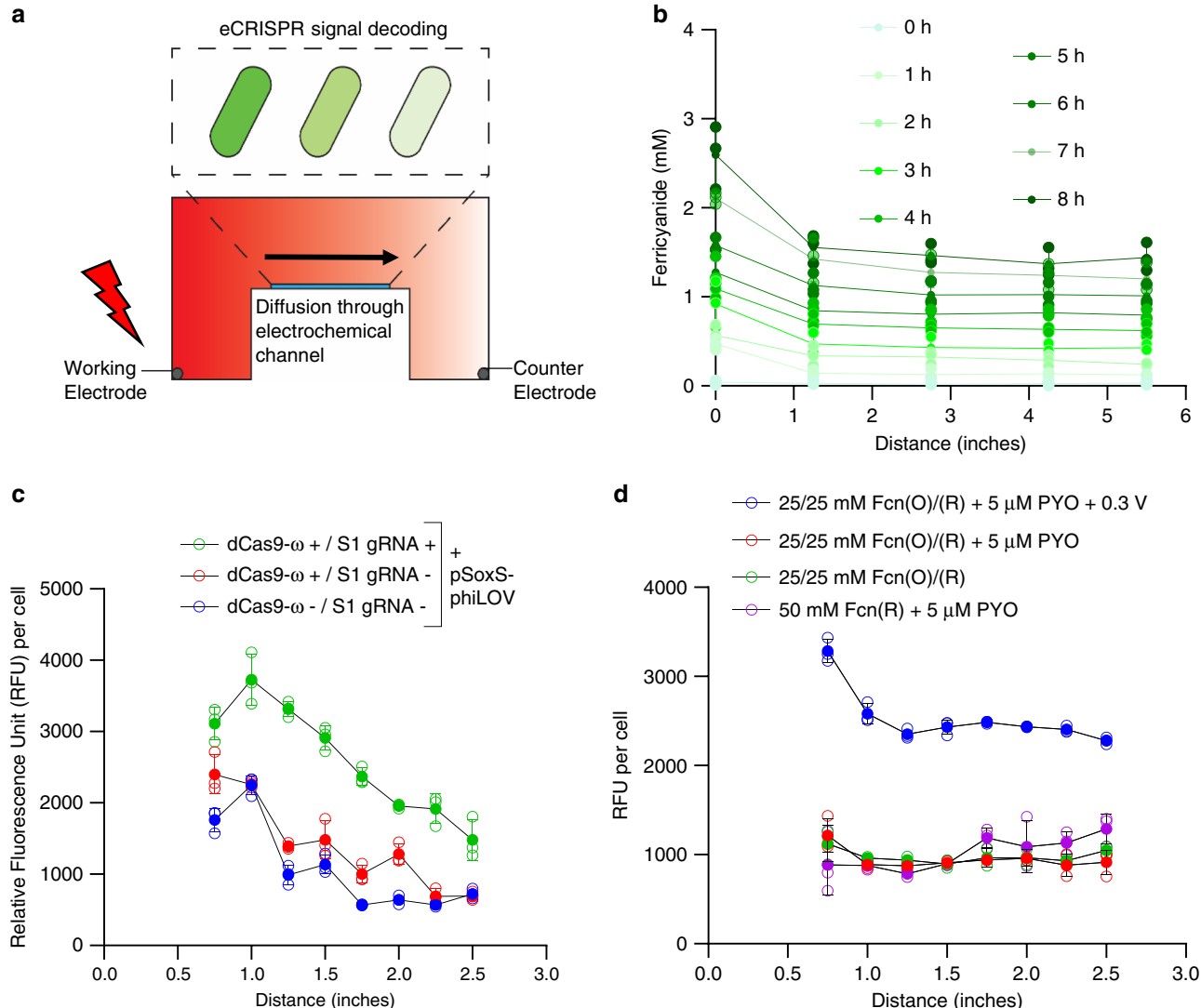

**Fig. 5 Spatiotemporal electronic control of eCRISPR. a** Scheme of the gel apparatus used to generate ferricyanide gradient upon electrical induction. An Owl easy cast gel electrophoresis system is filled with minimal-M9 media and supplemented with 50 mM Fcn(R). The left chamber is connected to the working electrode and the right chamber to the counter and both are connected to a potentiostat. Additionally, an Ag/AgCl is used as a reference electrode located in the left chamber. The entire apparatus is assembled and used inside an anaerobic chamber. To generate a gradient, an oxidizing potential of +0.3 V is provided across the gel apparatus. To study the effect of gradients on cells, cells are immobilized in gelatin hydrogels and cast onto microscope slides. Different cell populations are immobilized and the slides are placed at the center of the gel apparatus to expose the cells to the gradients. **b** Data indicating the formation of the ferricyanide gradient in the gel apparatus. An oxidizing potential of +0.3 V is applied across 8 h under anaerobic conditions and at each hour, aliquots are collected from various points in the gel apparatus and the amount of Fcn(O) present in the sample is measured via absorbance at 420 nm. To study the effect of Fcn(O) gradients on various cell populations, cells are assembled in gelatin hydrogels on slides and placed in the gel apparatus. After 8 h of electrical induction with +0.3 V, slides are removed and analyzed for phiLOV fluorescence using scanning laser confocal microscopy. Each data point consists of average fluorescence of 10,000 cells. **c, d** indicate cell fluorescence data from various experimental and control conditions noted in the text. In all figs, independent experimental replicates ($n = 3$) are represented by empty circles, mean indicated by filled circles and error bars indicate standard deviation. Source data for all figures is provided separately.

transcriptional control has been overcome in the recent past by exploiting various post translational approaches where dCas9 and transcriptional activators are engineered with dimerizing domains and expressed as two separate subunits that then dimerize upon optical[64] or small molecule[65] input. While we have successful integration of a CRISPRa system into our SoxS-mediated electrogenic circuits as indicated by ~15-fold activation of phiLOV, direct transcriptional activation of phiLOV from SoxS promoters was ~40-fold under similar plasmid copy number conditions[23]. Differences in efficiencies of activation in these cases could be attributed to contrasting ways in which transcription is initiated in these two scenarios. While in the case of the CRISPRa system,

the transcriptional activator ω subunit recruits and stabilizes RNA polymerase at sites upstream of desired promoters, in the case of SoxS promoters, the RNA polymerase is readily assembled on the promoter and upon SoxR oxidation there is a conformational change in −35 region of the SoxS promoter leading to transcriptional activation[66]. Interestingly, SoxS was recently shown as an activating subunit in CRISPRa system with better transcriptional activation over the ω subunit[34]. Overall, incorporation of CRISPR-based synthetic transcriptional regulators as an intermediate layer between electrogenetic promoters and genes of interest provides the flexibility to electrically target, activate and repress multiple genes simultaneously.

In conclusion, we propose an electrogenetic methodology in which direct connection between electronic signals and bacterial cells can mediate expression of target genes. Emergence of eCRISPR provides the capability to electronically target specific genes in the genome of organisms and importantly, the integration of CRISPR with electronics provides the capability to electrically turn ON and OFF several genes simultaneously. In this way, electronically programmed information is transmitted to and within biology using a medium of redox as a communication channel. As a proof of concept, we have demonstrated programed silencing of host defense responses and concomitant transgene activation. This builds on the multiplexed power of CRISPR. We believe that the further development of these capabilities to electronically target select genes across the host genomes could be a significant tool in bioelectronics research where so far, the focus has traditionally been on altering ionic currents and targeting cells and tissues rather than specific molecules and genes within cells. This work, therefore, furthers our ability to electronically control biological function. By altering voltage in a programmed manner, one can actuate genes in modified cells that, in turn, can mediate native biological signaling processes. We have exploited redox communication, in particular, as redox mediators provide a means for transferring electronic information from electrodes into biological systems. While not discussed in detail here, applications for such modulation are abundant. For example, quorum sensing is intrinsically connected to gut microbiome[67,68] and its ability to influence disease is well recognized. Integration of electrogenetic promoters with quorum sensing networks in bacteria could perhaps enable direct electric modulation of gut microbiomes, especially if these systems are incorporated into devices such as an ingestible electronic capsule[69]. This work will also broaden the application of synthetic biology, where problems pertaining to chassis compatibility are frequently reported. Simultaneous activation and repression mechanisms are important for moderating or supporting host behavior and for the successful adaptation of synthetic biology toolsets across a wide range of organisms.

## Methods

**Strains and plasmids**. Strains and plasmids used in this study are listed in Supplementary Table 1. Strain *E. coli* NB101 (Δ*rpoZ, lacZ*), created for use in CRISPR experiments was generated from background strain *E. coli* ZK126[70] using primers described here in ref. [33] and λ red recombinase[71]. All the primers used in building plasmids are listed in Supplementary Table 2 and we used standard molecular biology protocols such as restriction cloning, Gibson assembly and site directed mutagenesis for construction of plasmids.

**Media and growth conditions**. We used lysogeny broth (LB) for all experiments performed in aerobic conditions. We grew overnight cultures in LB media at 37 °C and the following day reinoculated cultures in fresh LB media at 1:100 ratio and used as per instructions in each specific experiment. For experiments performed under anaerobic conditions, we first grew cells to $OD_{600}$ 0.6 in LB media under aerobic conditions, washed and later resuspended in minimal-M9 media (1x M9 salts, 0.4% glucose, 0.2% casamino acids, 2 mM $MgSO_4$, 0.1 mM $CaCl_2$ and 100 mM MOPS) and used for further experiments. We created anaerobic conditions in an anaerobic chamber (Coy, MI) using a gas mixture comprising of 90% nitrogen, 5% carbon dioxide and 5% hydrogen. Ampicillin (50 μg/ml), kanamycin and chloramphenicol (each 25 μg/ml) were added as per requirements.

**Fluorescence measurements**. We used a Spectramax M2 plate reader (Molecular Devices, CA) to measure GFP fluorescence with excitation/emission wavelengths of 488/520 nm. To measure phiLOV fluorescence via flow cytometry (BD, NJ), we used constant forward and side scatter settings and a 488 nm laser with a 530/30 green filter (Supplementary Fig. 14). A minimum of 50,000 cells were used for each sample and data analysis was performed using FACSDiva (BD) and MS-Excel.

**AI-1 reporter assay**. We grew AI-1 reporter cells (an *E. coli* strain containing plasmid pAL105[72]) overnight in LB media with kanamycin and tetracycline at 37 °C and the next day diluted the reporter cells 2500x in fresh LB media. We also diluted conditioned media (CM) from AI-1 producer cells to 1000x or 100x in LB

media. We added 10 μL of each diluted CM sample with 90 μL of diluted reporter cells in FACS tubes (BD, NJ) and incubated for 3 h in a 30 °C shaker. After incubation, we measured luminescence using a GloMAX luminometer (Promega, WI). AI-1 activity was reported as the luminescence values normalized to a control with fresh LB media added instead of CM.

**Electrochemical set up for electrical activation**. To electrically activate the cells, we devised an electrochemical sample setup inside the anaerobic chamber that consisted of two glass vials with one vial consisting of 1 ml of cells in minimal-M9 media supplemented with Fcn(R) and PYO and the other consisting of 1 ml of Fcn (O). We used two gold wires (0.5 mm diameter, ~50 cm in length) immersed in each vial as working and counter electrodes with an Ag/AgCl electrode (CH Instruments, TX) as a reference electrode. Vials were connected by two salt bridges. We connected all electrodes to a CHI Instruments 600-series potentiostat for controlling voltages. For conversion of Fcn(R) to Fcn(O), we applied a constant oxidizing voltage via the working electrode with Ag/AgCl as reference. Since the voltage was kept constant, the amount of time taken to apply the desired electric charge varied from few seconds to 3–4 min. Upon application of desired charge in the anaerobic chamber, we removed the cells from vials, transferred them to culture tubes and incubated in a 37 °C incubator for varying amounts of time inside the anaerobic chamber without shaking. For time course experiments, we removed 100 μL of cells at each time point and fixed with 2% paraformaldehyde for a minimum of 15 min and used for flow cytometry.

**Spatiotemporal redox signaling apparatus**. We used an Owl Easycast B1A (Thermo Scientific) gel electrophoresis system in the anaerobic chamber to create electrochemical gradients. We degassed the minimal-M9 media by constant stirring under anaerobic conditions and added 200 ml of degassed media into the electrophoresis gel system. Later, we supplemented the media with 50 mM potassium ferrocyanide and 5 μM PYO. To create a gradient, our electrochemical setup consisted of a working electrode connected to the left chamber of the device, a counter electrode connected to the right chamber of the device, and an Ag/AgCl reference electrode placed near the working electrode in the left chamber of the device. Reference was isolated from the media by a low pass frit. All the electrodes were connected to a CHI 620E potentiostat and a desired electric charge was applied.

**Gelatin encapsulation of bacterial cells and imaging**. We grew cell populations in minimal-M9 media to $OD_{600}$ 0.5 and a 2x concentration of $OD_{600}$ 0.5 cells was mixed with a 2% gelatin solution maintained at 37 °C in minimal-M9 media. We poured this mixture evenly onto glass slides which were demarcated into 0.25 sq. inch segments and the mixtures were cooled for 10 min at 4 °C (Supplementary Fig. 13). Microscope slides with each of the three cell types were then transferred into an anaerobic chamber and carefully placed side-by-side in the middle chamber of the gel apparatus described above inside a 30 °C incubator in the anaerobic chamber. Slides were carefully placed with minimal residence in the media prior to the application of the electric potential. After application of electric potential for 8 h, the microscope slides were removed from the gel apparatus and analyzed by laser scanning confocal microscopy for phiLOV fluorescence (488 nm laser and 530/30 green filter) at the demarcated positions on the glass slides. Images were analyzed using ImageJ to determine the per-cell fluorescence of a minimum of 10,000 cells.

**Reporting summary**. Further information on research design is available in the Nature Research Reporting Summary linked to this article.

## Data availability
The data that support the findings of this study are available from the corresponding author upon request. The source data underlying Figs. 2–5 and Supplementary Figs. 1–12 are provided as a Source data file. All other relevant data, including plasmid sequences and plasmids, are additionally available upon request.

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

## Acknowledgements

We would like to thank Dr. Luciano Marraffini for gifting JEN201 strain and plasmids pWJ89, pWJ66 and pWJ108. We would like to thank Dr. Shirley Micallef for the *Salmonella enterica* serovar Typhimurium LT2 strain. We would also like thank DTRA (HDTRA1-19-0021), NSF (DMREF #1435957, ECCS#1807604, CBET#1805274), the National Institutes of Health (R21EB024102), and Agilent, Inc., for their generous support. All raw data are available in our Supplementary Information or upon request.

## Author contributions

N.B., E.V., and W.E.B. designed and performed experiments and wrote the papers. K.T.S. helped in performing quorum sensing experiments. P.H. helped in construction of strains and plasmids and general discussion. G.F.P. helped in overall discussion of paper.

## Competing interests

The authors declare no competing interests.
