## [Peer Review File · Nature Communications]

Reviewers' Comments:

Reviewer #1:

Remarks to the Author:

The manuscript by Bhokisham et al. describes the construction of eCRISPR, an electrogenetically controlled version of gene expression regulation via subsequent CRISPRa and CRISPRi. Utilizing the bacterial SoxS promoter which is activated during the oxidative stress response, the authors were able to make gRNA expression inducible by pyocyanin addition and ferricyanide-mediated application of a range of voltages. With this, Bhokisham et al. could demonstrate electrogenetic control of quorum sensing. While this study does represent an incremental advance in comparison to their earlier work, it lacks true novelty, has several weak points and lacks a clear application. Overall, this does not make this study eligible for publication in Nature Communications. Below, I list some of my concerns that the authors might want to address before their submission to a different journal.

Major points:

- 1) As already cited by the authors in the study, using the SoxS promoter for redox-induced gene expression in bacteria was already performed by the same research group. Therefore, the novelty of this work, which seems to repeat this by simply exchanging the reporter gene for gRNAs, needs to be drastically improved.
- 2) From the current study, the overall relevance is very unclear as no real application can be found in the paper. As the conclusion mentions microbiome applications of their technology, the authors should aim to demonstrate that their technology is superior to existing approaches for this application.
- 3) Both Fig. 1e and 2e show extremely low fold changes in their quorum sensing AI-1 reporter cells (lower than 2-fold for Fig. 2e). This makes the claim of the authors to control quorum sensing seem exaggerated and questions the practical relevance of their approach. The authors should aim to increase the fold-change of their technology.
- 4) Related to point 3, Fig. 1c shows substantial expression leakiness of the fluorescent output protein in the condition lacking inducer (pyocyanin) which is potentially the reason for the low fold-change. The authors should decrease this leakiness and investigate why there does not seem to be any leakiness in Fig. 2c which has a very similar set-up.
- 5) Experiments involving pyocyanin are simply small molecule-regulated gene expression experiments and have no direct connection with electrogenetics which makes them slightly misleading in this context. The focus should be on electrogenetics experiments using ferricyanide. The authors should also demonstrate why they do not directly upregulate LasI via electrogenetics but rather indirectly upregulate LasI by CRISPRa.
- 6) The authors mention approaches in mammalian cells using split CRISPR approaches which can be regulated by small molecules etc. There is no reason why these approaches should not work in bacteria and the authors should use these approaches to improve their eCRISPR fold-change.

Minor points:

- 1) The authors should evaluate what effects (viability, metabolism etc.) their knockdown of SoxS and the ensuing oxidative stress response has on their bacteria.
- 2) The authors should demonstrate that no other exogenous stimuli can activate the SoxS promoter to avoid unspecificity, especially in their intended microbiome environment.
- 3) There seem to be two 'Figure 1' figures in the manuscript. The authors should fix this.

- 4) The chosen terminology of information filter and information amplifier seems far-fetched / overblown considering that what happens is simply the downregulation of a couple of genes.
- 5) Fig. 3 and 4 are so similar that they should be combined into one figure.
- 6) In Supplementary Fig. 8, the order of the inducer concentrations after 0 should be reversed.
- 7) As far as I can see, there is no repression shown in Supplementary Fig. 10. The authors should clearly explain why they label this 'Simultaneous activation and repression'.

Reviewer #2:

Remarks to the Author:

Review of Bhokisham et al., "A redox-based electrogenetic CRISPR system to connect with and control biological information networks".

The manuscript develops and optimizes a set of genetic constructs to enable electronic control of gene expression in *E. coli* (and also *Salmonella enterica*). The study builds off of previous reports that demonstrated electronic control of gene expression mediated by SoxR with CRISPR based activation and repression of gene expression. The electronic control of signal exchange between cells was also demonstrated. Overall the manuscript was well written and clearly described well-designed and executed experiments. Many optimization strategies were tested and thoroughly reported throughout the main text and supplementary information, although the some findings could be clarified, see comments below. Whether the results of this study are of general interest to the readers of Nature is unclear. The findings are exciting: electronic control of gene expression, using CRISPR to regulate gene expression, and regulating quorum sensing via electronic inputs, however all of these findings have been reported previously, including a recent paper in Nature by many of the same authors. The combination of CRISPR gene regulation with the constructs developed in reference 22 are not a big enough advancement to warrant publication in Nature. The authors mentioned that the technologies reported may lead to "biohybrid microelectronic devices", and perhaps a manuscript focused on such devices or other applications of these constructs would greatly increase the novelty and impact of the findings. Below are more detailed comments about the manuscript.

1. The results presented references 22 (and 26, a duplicate reference) are similar, perhaps too similar, to the results of this manuscript. Two main findings of this manuscript are 1) the use of Fcn/Pyo to react with SoxR and drive gene expression in cells and 2) using such electrical control to regulate the exchange of quorum sensing signals. The major addition in the current manuscript is the incorporation of CRISPR components as a middle layer to regulate the activation and repression of gene expression. The CRISPRa and CRISPRi constructs also were developed previously (ref 33 and subsequent articles). Therefore although the synthetic biology components are assembled in a new way, the end result seems predictable. Figure 1 and 2 of the main text don't seem like noteworthy findings based on these previous studies.

2. The results related to Figure 3 I find very confusing. The ability to repress gene expression from the genome was clear, including the ability of both dCas9 and dCas9-w to repress. It was unclear why repressing soxS improved the expression from the soxS promoter on the plasmid. Was the idea that soxS and downstream genes reduced levels of oxidized pyocyanin (and maybe Fcn(O)too)? Did activation of the soxS pathway change the global state of the cell somehow indirectly influencing gene expression from a plasmid? Did soxS activation influence the activity or lifetime of activated SoxR? I reread this section a few times, and why deletion or repression of soxS influenced gene expression on the plasmid remains murky.

3. There are several key words/concepts used throughout the manuscript to connect electronic processes with biological processes that are hard to understand. On line 65 it is said that the "bandwidth" over which electronic information can be transmitted to biological systems has been expanded. The meaning of bandwidth here is unclear. The concepts of a filter and amplifier are mentioned in the abstract and on lines 110 and 366, but these seem more fanciful rather than help readers conceptualize the gene circuits developed in this manuscript. It reminds me of the use of a band-pass filter in work by Ostermeier in 2009, however in this previous case the electronics analogy actually helped readers to understand the concept of the gene circuit. I vaguely see how the reported constructs are filters and amplifiers, but these labels aren't helpful. Don't many circuits act as amplifiers? Perhaps these terms are trying to strengthen the connection between electronic and biological control, but the filter and amplifier functions are seemingly unconnected to the use of electronic inputs.

4. Similar to the use of electronics language, I am not sure the use of information processing and information networks is helpful. It is fine to call gene regulation and responding to external cue information processing, but especially "information networks" in the title seems unneeded and perhaps misleading. No metrics for "information" are used anywhere in the paper.

5. On line 457 it states "as redox mediators provide a means for transferring electronic information from electrodes into biological systems and the reverse". This idea of information exchange back and forth between electronics and biology is also mentioned on line 57. The results presented do not seem to demonstrate information transfer from biology to electronics, so the authors should avoid overstating what was demonstrated in this manuscript.

6. The relatively poor upregulation of quorum sensing activity shown in Fig. 2E is confusing. The explanation given in the text is given starting on Line 273. So the problem is under anaerobic conditions signal accumulation is reduced, -0.1 C is not enough charge, and 2 hours is not long enough to see a strong response? The 2 hour problem is fixable. It also seems applying more than -0.1 C is possible, right? Sounds like these experiments should be run again given the results of Fig. 2C. Why is relative activity more than 2 for 0 C? This is relative to the addition of no signal or supernatant from a strain that doesn't make signal?

Minor concerns:

7. The manuscript does not point out that pyocyanin can be toxic to cells and generates reactive oxygen species (well line 290 points out ROS capabilities but it is still not clear if the cell is strongly influenced by pyocyanin at the concentrations used). Looking quickly, I saw the MIC of pyocyanin against bacteria to be about 100-300 microM, which is an order of magnitude larger than concentrations used in the study. However, 10 fold less than the MIC might still have an observable influence on cell growth and behavior, particularly given pyocyanin redox activity. I am not necessarily thinking about *E. coli*, but other cells that might be more sensitive to redox stress. This study, 10.3390/toxins8080236, reported the toxicity of pyocyanin on human cells in the range of 10-100 microM. This certainly could limit context in which pyocyanin is an inducer for gene expression. The findings reported in the manuscript are still noteworthy, but these potential limitations should be noted. No additional experiments would be needed to address this minor concern.

8. PAM is not explained in the manuscript. Defining the importance of the PAM site within the text is probably needed for the general audience of Nature. Likewise an explanation for adding the ribozyme sequences could be helpful.

9. The strain and plasmid names are hard to follow in some figures. The worst case is probably Fig. 3D (NB101+pNB01+dCas-w vs. NB101+ptt01+pNB02+dCas9-w, etc.). Is there a simpler way to understand these results than listing each plasmid and host? Perhaps delta-soxS, no repression of soxS, S1 production, soxS regulated S1 production or something similar. I understand wanting

to be exact about what was measured, but perhaps these details in the text or in the figure caption instead of the legend will make the paper easier to read.

10. The first line of the abstract mentions "rapid and programmable information transfer between electronics and biology", but the responses shown appear to take hours and are likely of similar speed to adding a chemical inducer. The rate of information transfer and response is a very relevant topic here, so perhaps some attention could be given to this issue in the discussion. How could information exchange rates be increased?

11. The use of "etc." several times on page 3 was distracting. If you state such as and list several things etc. does not seem needed.

12. Designing genetic constructs for use in multiple chassis is a good goal for synthetic biology, and it does seem that modular tools such as CRISPRa and CRISPRi will contribute to such goals. Showing that the construct works in *Salmonella* is worthwhile to report, but it should be noted in the discussion that *E. coli* and *Salmonella* are closely related species of bacteria. Although *Salmonella* is a "non-chassis strain", it is not too surprising constructs in *E. coli* would be functional in *Salmonella*. The results do not suggest the ability to easily move these constructs into other species of bacteria.

Reviewer #3:

Remarks to the Author:

Summary:

This manuscript describes the expansion and improvement of a system previously developed by the Bentley lab for electrode-based control of gene expression.¹ The original system used a redox-sensitive SoxS promoter (regulated by the SoxR repressor) to control expression of a fluorescent protein via charge transfer from an electrode. Charge transfer is mediated by two small molecules, ferricyanide (Fcn) and pyocyanin (PYO). The major changes made here are to 1) add an additional layer of control by using the SoxS promoter to control expression of a CRISPR-based transcriptional activation, 2) enhance the dynamic range of the eCRISPR system by repressing the native SoxS response, and 3) transfer the eCRISPR system to *Salmonella enterica*. Overall, the eCRISPR system worked to induce expression in *E. coli* and *S. enterica*, although in *S. enterica*, electrical induction worked better without the added layer of CRISPRa control. This work represents an important next step in the development of electrode-controlled induction systems, but the manuscript would benefit from additional explanation of the overall engineering strategy, high level interpretation of results, and a brief, but clear outline of the electrode induction method in the results.

Major comments:

1. Overall, I found the manuscript somewhat difficult to follow. Addition of some explanatory statements throughout, and more descriptive plasmid and strain names would be helpful. A few examples:
 - a. It would be helpful at line 345 to state that the location of the gRNA binding site is what dictates whether dCas9- ω acts as an activator or repressor (i.e., binding in the promoter leads to CRISPRa, while binding after the TSS leads to CRISPRi).
 - b. Plasmid pTT01 could be referred to as pSoxS-phiLOV, similar to the pMC-phiLOV. Overall, replacing alphanumeric codes with more meaningful abbreviations would be useful throughout the manuscript.
 - c. Lines 184-186: A more detailed explanation of what is the seed region of the spacer and why point mutations would be expected to tune CRISPRi would be helpful.
 - d. Lines 293-295: At this point it would be helpful to clarify that repression of SoxS on the genome should not directly influence regulation of the SoxS promoters in the engineered constructs, because SoxR expression is unaffected.

2. Why are both FCN and PYO necessary for activation? PYO can be re-oxidized abiotically at electrodes, so it is not clear why the additional mediator is needed. This may have been explained in the previous paper, but it should be briefly recapped here.
3. Lines 148-149 indicate that cells were transferred back to aerobic conditions after charge transfer. Why was this done, and how long were cells incubated? If incubation times in figures refer to this post-charge transfer incubation, then the amount of time it took to do the charge transfer should also be noted. Overall, I think a clear description and diagram in the results section of the overall workflow for charge transfer experiments would be helpful.
4. Lines 273-279: If the 2-hr incubation and small amounts of charge transfer did not lead to strong induction of AI-1 production, why were these conditions chosen? A better explanation is needed for why experiments with the AI-1 output were done differently than experiments with the Philov output.
5. The result in Figure 2E seems marginal. Significance testing may be appropriate here, it would be good to consult with a statistician. Also, please discuss potential reasons for the high leaky expression in the AI-1 system compared with the Philov system. It seems like the 0 charge transfer condition is well above the no AI-1 background.
6. Lines 367-368: The analogy to a filter and amplifier are not clearly explained. Please state more clearly what is the input and what is the output in each analogy. It would also be helpful to explain how the filter and amplifier functions developed here could be integrated into larger engineered circuits.
7. Lines 458-467: Application of this system to gut microbiomes seems very far in the future. Are there any medium-term applications that could also be discussed here? Nearer-term application of eCRISPR in a biotech setting seems more feasible.
8. The methods do not provide sufficient information about how charge transfer was controlled. If a constant voltage was applied for a pre-determined amount of time, how were different charge transfer amounts achieved? This relates back to point 3.
9. For figure S10, why was a combination of Fcn and PYO used, instead of just PYO? Again, this is where an explanation of the specific roles of Fcn and PYO in the system would be helpful.

Minor comments:

1. All potentials noted in the manuscript should indicate what they are relative to, i.e., is it 0.5 VSHE or 0.5 VAg/AgCl?
2. There are a few places in the manuscript where symbols were replaced by blocks: e.g., Line 270, Line 397.
3. There are inconsistencies throughout the manuscript whether there is a space between a number and a unit.
4. Line 339: The text indicates that DJ901 is a Δ soxRS strain, but Figure 3 indicates that it is a Δ soxS strain. Please correct.
5. Line 414: change 'be' to 'by'
6. Line 434: remove 'While'
7. Line 444-445: I recommend softening the language here. I'm not sure that use of two redox mediators should be considered 'direct connection between electronic signals and bacterial cells'.

References

(1) Tschirhart, T.; Kim, E.; McKay, R.; Ueda, H.; Wu, H.-C.; Pottash, A. E.; Zargar, A.; Negrete, A.; Shiloach, J.; Payne, G. F. Electronic Control of Gene Expression and Cell Behaviour in *Escherichia Coli* through Redox Signalling. *Nat. Commun.* 2017, 8, 14030.

Reviewers' comments:

Reviewer #1 (Remarks to the Author):

The manuscript by Bhokisham et al. describes the construction of eCRISPR, an electrogenetically controlled version of gene expression regulation via subsequent CRISPRa and CRISPRi. Utilizing the bacterial SoxS promoter which is activated during the oxidative stress response, the authors were able to make gRNA expression inducible by pyocyanin addition and ferricyanide-mediated application of a range of voltages. With this, Bhokisham et al. could demonstrate electrogenetic control of quorum sensing. While this study does represent an incremental advance in comparison to their earlier work, it lacks true novelty, has several weak points and lacks a clear application. Overall, this does not make this study eligible for publication in Nature Communications. Below, I list some of my concerns that the authors might want to address before their submission to a different journal.

We thank the reviewer for the comments. We appreciated the reviewer's concerns regarding lack of novelty and application and we have tried to address these concerns by the addition of new experiments detailed in the manuscript as well as in our responses to the specific concerns below.

Major points:

1) As already cited by the authors in the study, using the SoxS promoter for redox-induced gene expression in bacteria was already performed by the same research group. Therefore, the novelty of this work, which seems to repeat this by simply exchanging the reporter gene for gRNAs, needs to be drastically improved.

The reviewer is correct in noting the swapping of a reporter gene for gRNA is one of the central differences between this work and our previous efforts. We would like to suggest, however, that the true novelty is shown first by incorporating CRISPR and then demonstrating enhanced control and meaning that accompanies its inclusion. That is, CRISPR enables generic application, multiplexed function, and tremendous simplification for future application. In our previous work, a mutated host was needed to show a significant response. Here, the concomitant downregulation of the oxidative stress response served to amplify the otherwise upregulated genes-of-interest. Such coordinated control is not feasible otherwise. Further, we show that essentially the same controller works in different species. That we demonstrate this cassette or "drop in" type of utility is also quite novel and significant. Perhaps most importantly, we provide new data to clearly demonstrate that electronic actuation is significantly different than simple addition of a chemical inducer to suspended cells – the principal inducing modality in our previous work. In the current work, we show how CRISPR functions as an information processor by attenuating the otherwise upregulated stress response. We show spatially resolved actuation, enabled by electronics. This new manuscript highlights the strength of electronics and hopefully sheds new light on the importance of electronic control. We note that all reviewers expressed similar concerns relative to our weak inclusion of information processing utility. We have significantly altered the manuscript to emphasize the strength of electronic actuation. In this context, the advantage of CRISPR and the new noise reducing functionality of CRISPR should be more clear.

That is, from an information processing perspective, we suggest the oxidative stress response to electrically imposed redox signals as 'genomic noise' that clouds the information transfer into the biological system. The suppression or 'filtering' of this genomic noise leads to increased signal strength (or 'signal amplification'), in this case the increased upregulation of genes of interest. The resulting consequence of this coordinated signal filter and amplifier is that the electrically stimulated populations better respond to the electric signals.

We have added a new Fig. 4 where electrical activation of biological populations was performed amid biochemical gradients that were generated by electrical signals. Populations engineered with CRISPR circuits displayed responses correlating directly with biochemical gradients. Populations without CRISPR circuits were not aligned with the gradients and, in general, displayed lower responses (or, higher elevated signal error). In this way, the addition of CRISPR circuits resulted in an overall improvement of information transfer.

2) From the current study, the overall relevance is very unclear as no real application can be found in the paper. As the conclusion mentions microbiome applications of their technology, the authors should aim to demonstrate that their technology is superior to existing approaches for this application.

We readily admit that applications of these methodologies are going to appear in the future. We suggest they will result in areas where biological signalling is central to understanding biological function, notably from the perspective of controlling noisy environments. We had previously suggested the microbiome as an area where applications will emerge. The molecular signalling at play in the microbiome is currently inaccessible. The oxygen levels, for example, change from fully oxygenated within the gut epithelia to strictly anaerobic in the intestinal lumen – all in the space of a few millimetres. Our electronic activation of gene expression will surely find utility in *in vitro* studies (e.g., gut on a chip) where local redox gradients are significant and intentionally (or otherwise) generated. This ability will help to ensure faithful recapitulation of the native environments. Additionally, there are many companies pursuing engineered commensal bacteria as a means to treat intestinal disease. Their design is facilitated by *in vitro* models where microenvironments can be controlled. There are also companies building “smart” pills that interrogate chemical and physical signatures as they pass through the gastrointestinal tract. While a long way out, engineered cells that respond to localized redox signals in the GI tract could be activated by a passing electronic pill. Such technology-assist concepts are under currently under development in several laboratories, including ours.

Similar redox-based molecular communication exists in such disparate environments as the skin, brain and the rhizosphere. A cell-based electronic CRISPR placed in these environments (including in labs attempting to recapitulate the natural settings), will greatly enhance our understanding of redox communication in complex settings. Additionally, such a means to electronically control cells will impact fields such as metabolic engineering and synthetic biology where the production of chemicals and biologicals needs to be carefully controlled. Finally, we anticipate that many investigators will be inspired by this work and they too will be interested in developing applications. We highlight two papers below that have emerged since our original submission. In sum, we do not believe applications that will employ electronic activation and control of gene expression are far away.

3) Both Fig. 1e and 2e show extremely low fold changes in their quorum sensing AI-1 reporter cells (lower than 2-fold for Fig. 2e). This makes the claim of the authors to control quorum sensing seem exaggerated and questions the practical relevance of their approach. The authors should aim to increase the fold-change of their technology.

We have revised these sections to more accurately convey the significance of these specific data and their importance. This issue was similarly raised by several reviewers and we feel this is a consequence of our previous explanation, not the significance of the data. At the outset, the AI-1 generation in the stimulated cells was not shut down in the absence of charge. This is most likely due to readthrough transcription and translation of the AI-1 synthase and the fact that the AI-1 synthase, being an enzyme, will continuously produce AI-1 throughout the extended induction period. That is, we readily agree that the fold changes in AI-1 activity were low. While we show ~15-fold increase in phiLOV expression and this demonstrates the ability to generate a large fold change in actuated cells, the fold changes in AI-1 synthesis were not as impressive. Several factors additional contribute:

First, while phiLOV fluorescence is directly measured by FACS, AI-1 activity is not. AI-1 activity is determined by the bioluminescence obtained from AI-1 reporter cells and several factors such as the linearity and responsiveness of the AI-1 reporter cells to AI-1 could directly affect the AI-1 activity.

Second, the various metabolic limitations inside the electronically-stimulated cells (that govern the AI-1 production) could play a role in the amount of AI-1 produced by LasI. That is, there necessarily needn't be a paucity of LasI, but a concomitant cofactor or substrate for LasI that could be limiting. Both conditions are not necessarily controlled by CRISPR.

Finally, and more importantly, in certain systems minor increases (say a 2-fold increase) in signal input is enough to activate 100% of cells downstream or make a significant impact on downstream processes. For example, relatively low levels of gRNA might be enough to silence or activate genes downstream - expressing several fold more gRNA might not necessarily make a difference. It might otherwise waste energy (e.g., suppose 2-fold increase in AI-1 is all that is needed and a squarely linear response to a signal is not obtained). Also, for cases where significant amplification is needed, we have previously shown that by exogenously activating T7 polymerase and by including a downstream T7 promoter, significant amplification can be achieved (Tsao et al, (2010)).¹ That said, in this specific case, we have modified the importance of the quorum sensing data in Figures 1D and 2D. Here, the controls and low charge cases are grouped together to show a set of conditions where QS-responsive cells are already “communicating” and the high charge cases are lumped to show the stimulation of this QS activity. Note, it is important to convey that this is presented in the context of molecular communication. The electrogenetically activated cells parse the redox activity that is electronically programmed. These cells then make AI-1 and this is conveyed to other cells. In Scheme 1, we suggest that this process will generate the highest signal nearest the electrode. The redox signal will be “read” by CRISPR containing cells near an electrode. These, in turn, will make AI-1 that is subsequently transmitted as a function of time and distance to locales more distant. That a 2-fold increase is generated only serves to modulate the signal which is ultimately conveyed over time. The electronically programmed information is then biologically transferred using biological signalling to responding cells. We have modified Figures 1 and 2 to reflect this revised interpretation. We have also generated new results to specifically highlight this information processing feature in a new Figure 4.

4) Related to point 3, Fig. 1c shows substantial expression leakiness of the fluorescent output protein in the condition lacking inducer (pyocyanin) which is potentially the reason for the low fold-change. The authors should decrease this leakiness and investigate why there does not seem to be any leakiness in Fig. 2c which has a very similar set-up.

We thank the reviewer for pointing this question and while it does seem the conditions used in Fig. 1C and 2C are similar, the conditions are indeed quite different.

The substantial leakiness in Fig. 1C is due to the leakiness of the SoxS promoter expressing the CRISPRa gRNA. The SoxS promoter is inherently leaky under aerobic conditions because oxygen recycles PYO from reduced to oxidized state and this oxidized PYO can continuously drive gene expression from the SoxS promoter. Now, when a gRNA is expressed from the SoxS promoter, gRNA in combination with dCas9 can form stable complexes (half-life for ~6 hours) and these complexes can drive gene expression via CRISPRa. Hence, the leaky expression is enhanced in Fig. 1C.

However, the experimental set up in Fig. 2C is quite different. Under anaerobic conditions, there is no oxygen to oxidize the PYO and hence there is very little leaky expression from the SoxS promoter. Under anaerobic conditions, recycling of PYO is enabled by the oxidized ferricyanide which, in turn, is provided by electric signals. Hence without electrical induction, leaky expression in Fig. 2C is minimal.

5) Experiments involving pyocyanin are simply small molecule-regulated gene expression experiments and have no direct connection with electrogenetics which makes them slightly misleading in this context. The focus should be on electrogenetics experiments using ferricyanide. The authors should also demonstrate why they do not directly upregulate LasI via electrogenetics but rather indirectly upregulate LasI by CRISPRa.

Pyocyanin plays an intrinsic and direct role in electrical induction and without pyocyanin there is no electrical activation from the SoxS promoter. Pyocyanin is oxidized by the electrode and directly interacts with SoxR regulator that drives the SoxS promoter. Ferricyanide serves to amplify the initial levels of gene expression stimulated by pyocyanin². Results shown in Fig. 4D, where control experiments are performed without PYO, also support this argument.

Additionally, small molecule experiments with just PYO are performed under aerobic conditions and these experiments are predictive of conditions (e.g., such as concentrations of PYO) to be used for electrical induction experiments under anaerobic conditions. Hence, initial characterization studies are done under aerobic conditions using PYO alone. Later, to display electrical induction under anaerobic conditions, Fcn (O) is directly added along with PYO. Once concentrations and conditions are optimized, Fcn (R) is added along with PYO and electric charge is provided under anaerobic conditions. Thus, we followed a step-by-step optimization of each condition prior to final electrical activation. In the electronic system, PYO and

Fcn are continually modified over time, both by the cells and by the electrode, while the small molecule experiments enable precise attribution to pulse additions of small molecule actuators.

Relative to the second part of this concern, the data in Figs. 3 and 4 were performed wherein genes of interest were directly expressed from SoxS promoters. The direct expression of LasI from SoxS was shown by our group earlier. In the current study, we show the inclusion and propagation of cellular computation...additional layers of regulation between the SoxS promoter and LasI gene expression are important. We did not adequately describe this in the original manuscript. We envisaged CRISPR to be this flexible regulatory layer that can perform diverse functions (both CRISPRa and CRISPRi) and our attempts to do this are now detailed in Fig. 3 and Supplementary Fig. 13.

6) The authors mention approaches in mammalian cells using split CRISPR approaches which can be regulated by small molecules etc. There is no reason why these approaches should not work in bacteria and the authors should use these approaches to improve their eCRISPR fold-change.

We allude to the use of small molecule-based approaches to control expression of Cas9 and dCas9 enzymes because direct transcriptional control of these enzymes have been proven to be difficult. We do not foresee major improvements in fold change in eCRISPR using a small molecule-based approach.

Another way to increase CRISPRa would be to engineer new synthetic promoters for bacterial CRISPRa. One such study of a synthetic pSoxS promoter with SoxS as the activating domain was shown to improve CRISPRa fold change³. However, whether this system can be integrated into our eCRISPR concept would need to be studied further as SoxS is involved in both these systems.

Minor points:

1) The authors should evaluate what effects (viability, metabolism etc.) their knockdown of SoxS and the ensuing oxidative stress response has on their bacteria.

The effects of presence/absence of SoxS is fairly well established in the literature. This is one of the key reasons we have elected to use SoxS in these studies. Elevated SoxS has a direct role in four major aspects of metabolism including detoxification, DNA damage protection and repair, methionine and aromatic amino acid synthesis and cell wall synthesis. A few of the genes that are directly affected by SoxS include:

1. *sodA* coding for superoxide dismutase that removes superoxide radicals from the cell.⁴
2. *Zwf* coding for glucose 6-phosphate 1-dehydrogenase to increase NADPH levels and antioxidant defence.⁴
3. *acnA* coding for aconitase A involved in the TCA cycle and more resistant to oxidation than its isoenzyme aconitase B.⁵
4. *fumC* coding for fumarase C in the TCA cycle and more resistant to oxidation than its isoenzyme FumA and Fum B.⁶
5. *lpxC* coding for UDP-3-O-acyl-N-acetylglucosamine deacetylase that catalyses lipid A biosynthesis.⁷

6. *fur* coding for Fe²⁺ uptake regulator Fur for repression of Fe²⁺ uptake during oxidative stress.⁸

These genes all play a vital role in mediation of oxidative stress defense responses. In our other studies, we have provided data on the metabolic activity of the electrode-actuated cells². Metabolism was significantly affected. The yield of acetate per glucose uptake, for example, was significantly altered. Again, we note that the downregulation of SoxS regulated genes was taken as an objective here because it enabled the programmed cells to focus their response to the genes-of-interest.

- 2) The authors should demonstrate that no other exogenous stimuli can activate the SoxS promoter to avoid unspecificity, especially in their intended microbiome environment.

The SoxS promoter is known to be activated by redox cycling compounds such as pyocyanin and paraquat (as shown in our study), phenazine methosulphate, plumbagin, nitric oxide and super oxide radicals (O₂⁻)⁹. In the gut environment, the two chemicals that can activate SoxS promoters are hydrogen peroxide and nitric oxide. While hydrogen peroxide orthogonally reacts with OxyRS system¹⁰, nitric oxide can stimulate SoxS. However, to continuously drive gene expression from the SoxS promoter, all these chemicals need oxygen for electron recycling. Under anaerobic conditions, Fcn or other electron acceptors are required to mediate electron-recycling thus making our electrical activation signals very specific for gene expression from the SoxS promoter. Under anaerobic conditions, PYO alone is not sufficient to activate SoxS promoters, as shown in our earlier work² as well as in Fig. 4 in this manuscript.

- 3) There seem to be two 'Figure 1' figures in the manuscript. The authors should fix this.

We apologize for the mistake. This has been corrected. We have a Scheme and Figure 1.

- 4) The chosen terminology of information filter and information amplifier seems far-fetched / overblown considering that what happens is simply the downregulation of a couple of genes.

We appreciate the reviewer's comment. Our previous manuscript did not adequately convey the importance of communicating between electronics and biology. Beyond ionic modalities, there are no other methods that have been shown to specifically stimulate gene expression via electronics. We have carefully executed this work by employing redox reactions. Redox activity is fundamental and widespread in biology and gaining electronic access to redox, we believe, will have transformative implications. We have significantly strengthened the information and communication aspects of our work in the new manuscript. For example, the introductory Scheme is improved, and we have added new data in Figure 4 to demonstrate the importance and the distinction.

- 5) Fig. 3 and 4 are so similar that they should be combined into one figure.

We thank the reviewer for the suggestion, and we have combined figures.

6) In Supplementary Fig. 8, the order of the inducer concentrations after 0 should be reversed.

We apologize for the mistake; the concentrations are reversed.

7) As far as I can see, there is no repression shown in Supplementary Fig. 10. The authors should clearly explain why they label this 'Simultaneous activation and repression'.

We have added the qPCR data indicating repression of SoxS using a SoxS specific CRISPRi gRNA. This gRNA was co-expressed with CRISPRa gRNA and both these co-expressed gRNAs are activated via ribozyme mediated post transcriptional processing.

Reviewer #2 (Remarks to the Author):

Review of Bhokisham et al., “A redox-based electrogenetic CRISPR system to connect with and control biological information networks”.

The manuscript develops and optimizes a set of genetic constructs to enable electronic control of gene expression in *E. coli* (and also *Salmonella enterica*). The study builds off of previous reports that demonstrated electronic control of gene expression mediated by SoxR with CRISPR based activation and repression of gene expression. The electronic control of signal exchange between cells was also demonstrated. Overall the manuscript was well written and clearly described well-designed and executed experiments. Many optimization strategies were tested and thoroughly reported throughout the main text and supplementary information, although the some findings could be clarified, see comments below. Whether the results of this study are of general interest to the readers of Nature is unclear. The findings are exciting: electronic control of gene expression, using CRISPR to regulate gene expression, and regulating quorum sensing via electronic inputs, however all of these findings have been reported previously, including a recent paper in Nature by many of the same authors. The combination of CRISPR gene regulation with the constructs developed in reference 22 are not a big enough advancement to warrant publication in Nature. The authors mentioned that the technologies reported may lead to “biohybrid microelectronic devices”, and perhaps a manuscript focused on such devices or other applications of these constructs would greatly increase the novelty and impact of the findings.

We thank the reviewer for his/her insightful comments. Heeding to the advice of the reviewer, we have added a new fig. 4 and emphasized more the information processing capabilities of the CRISPR and its significance in the bio-device environments. Interestingly, between our previous submission and now, a few more papers have emerged alluding to the importance of these efforts, one in *Nature Chemistry*¹¹ and one that credits our earlier work in *Nature Communications*² in defining an emerging field of “electrogenetics”¹². We respectfully disagree with the reviewer relative to the importance of this new work.

Below are more detailed comments about the manuscript.

1. The results presented references 22 (and 26, a duplicate reference) are similar, perhaps too similar, to the results of this manuscript. Two main findings of this manuscript are 1) the use of Fcn/Pyo to react with SoxR and drive gene expression in cells and 2) using such electrical control to regulate the exchange of quorum sensing signals. The major addition in the current manuscript is the incorporation of CRISPR components as a middle layer to regulate the activation and repression of gene expression. The CRISPRa and CRISPRi constructs also were developed previously (ref 33 and subsequent articles). Therefore although the synthetic biology components are assembled in a new way, the end result seems predictable. Figure 1 and 2 of the main text don't seem like noteworthy findings based on these previous studies.

From a synthetic biology point of view, the significance of this work is that while there are prior reports of CRISPRa, they are either constitutive or inducible but

not tunable. As far as we know, we are the first to report on tunable CRISPRa in bacteria (Fig. 1) and we have integrated this tunable CRISPRa with our electrogenetic promoters system (Fig. 2).

More importantly, we feel that the significance extends beyond synthetic biology and into information theory where the use of CRISPR results in tuning or reduction of genomic 'noise' and the resulting concomitant increase in desired signal output. A direct outcome of this is displayed in Fig. 4 where in a spatiotemporally controlled signal environment such as that at an electronic-bio device interface, populations embedded with CRISPR circuits display more aligned biological response to external signal gradients.

2. The results related to Figure 3 I find very confusing. The ability to repress gene expression from the genome was clear, including the ability of both dCas9 and dCas9-w to repress. It was unclear why repressing soxS improved the expression from the soxS promoter on the plasmid. Was the idea that soxS and downstream genes reduced levels of oxidized pyocyanin (and maybe Fcn(O)too)? Did activation of the soxS pathway change the global state of the cell somehow indirectly influencing gene expression from a plasmid? Did soxS activation influence the activity or lifetime of activated SoxR? I reread this section a few times, and why deletion or repression of soxS influenced gene expression on the plasmid remains murky.

We hypothesize that with the global oxidative stress defense response comprising ~20 genes being a huge metabolic burden to the cell and by merely suppressing this response, we could divert some energy towards expressing proteins of interest resulting in 4-fold increase in protein expression. This is a new attribute assigned to CRISPR and extends beyond simple multiplexing capability. The multiplexed capability is aligned with consequences in metabolic engineering. The transient downregulation of seemingly pleiotropic factors has been in the literature for quite some time, but to the best of our knowledge never shown relative to CRISPR or the oxidative stress response. Our group, many years ago showed an analogous methodology targeting downregulation of heat shock transcription factor, sigma32 using antisense RNA^{13,14} that enabled more yield of a desired product. We have included this reference in the revised version as well as several other examples from other groups. We readily acknowledge that we cannot rule out the possibility that effects of repression might have increased the lifetime of PYO in an oxidized state or towards increased recycling of PYO, but this too is an argument for the use the concomitant CRISPRi function. We agree with the reviewer that further experiment on the mechanisms of intracellular redox cycling would help to elucidate, but this is beyond the current scope.

3. There are several key words/concepts used throughout the manuscript to connect electronic processes with biological processes that are hard understand. On line 65 it is said that the "bandwidth" over which electronic information can be transmitted to biological systems has been expanded. The meaning of bandwidth here is unclear. The concepts of a filter and amplifier are mentioned in the abstract and on lines 110 and 366, but these seem more fanciful rather than help readers conceptualize the

gene circuits developed in this manuscript. It reminds me of the use of band-pass filter in work by Ostermeier in 2009, however in this previous case the electronics analogy actually helped readers to understand the concept of the gene circuit. I vaguely see how the reported constructs are filters and amplifiers, but these labels aren't helpful. Don't many circuits act as amplifiers? Perhaps these terms are trying to strengthen the connection between electronic and biological control, but the filter and amplifier functions are seemingly unconnected to the use of electronic inputs.

In this new manuscript, we have improved the connection between electronic signalling and its potential to actuate and coordinate biological responses. By "bandwidth", we suggest that by expressing CRISPR we have added a layer of cellular regulation that is concomitantly electronically actuated. This significantly enhances our ability to electronically control biological function in that the same electronic signal imparts more widespread biological function. This additional function is new to CRISPR and naturally new to electronic actuation. Here, we have targeted is the oxidative stress defense response that acts as a signal reduction agent or 'noise'; by suppressing or 'filtering' out this genomic noise, we have achieved an increase in signal or 'signal amplification'. The reviewer is correct in pointing out that by expressing say a T7 polymerase from an electrogenetic promoter results would likely demonstrate original signal amplification but here we are achieving a similar end result by suppressing the genomic noise, a practice that is employed widely in signal processing world.

4. Similar to the use of electronics language, I am not sure the use of information processing and information networks is helpful. It is fine to call gene regulation and responding to external cue information processing, but especially "information networks" in the title seems unneeded and perhaps misleading. No metrics for "information" are used anywhere in the paper.

We thank the reviewer for the comment. We have clarified a bit more on this point and restructured the paper on the topics of information processing aspects and why they are important.

5. On line 457 it states "as redox mediators provide a means for transferring electronic information from electrodes into biological systems and the reverse". This idea of information exchange back and forth between electronics and biology is also mentioned on line 57. The results presented do not seem to demonstrate information transfer from biology to electronics, so the authors should avoid overstating what was demonstrated in this manuscript.

We thank the reviewer for the comment and we agree that we do not have the element of communication from biology to electronics in this work. Frankly, the biology to electronics is more simple and we have several previous reports of this and have included them in our revised version.

6. The relatively poor upregulation of quorum sensing activity shown in Fig. 2E is confusing. The explanation given in the text is given starting on Line 273. So the problem is under anaerobic conditions signal accumulation is reduced, -0.1 C is not enough charge, and 2 hours is not long enough to see a strong response? The 2 hour problem is fixable. It also seems applying more than -0.1 C is possible, right?

Sounds like these experiments should be run again given the results of Fig. 2C. Why is relative activity more than 2 for 0 C? This is relative to the addition of no signal or supernatant from a strain that doesn't make signal?

The reviewer points to a topic also noted by the previous reviewer (above) that the fold changes in AI-1 activity are low. We refer the reviewer to our earlier response.

Minor concerns:

7. The manuscript does not point out that pyocyanin can be toxic to cells and generates reactive oxygen species (well line 290 points out ROS capabilities but it is still not clear if the cell is strongly influenced by pyocyanin at the concentrations used). Looking quickly, I saw the MIC of pyocyanin against bacteria to be about 100-300 microM, which is an order of magnitude larger than concentrations used in the study. However, 10 fold less than the MIC might still have an observable influence on cell growth and behavior, particularly given pyocyanin redox activity. I am not necessarily thinking about E. coli, but other cells that might be more sensitive to redox stress. This study, 10.3390/toxins8080236, reported the toxicity of pyocyanin on human cells in the range of 10-100 microM. This certainly could limit context in which pyocyanin is an inducer for gene expression. The findings reported in the manuscript are still noteworthy, but these potential limitations should be noted. No additional experiments would be needed to address this minor concern.

We thank the reviewer for the concern. Our group has performed some preliminary experiments on the effect of PYO on bacterial cell viability and we observed that at concentrations below 50 µM PYO, the cell viability was as high as ~95% even after 6 hours of incubation; there was no major difference with the no treatment samples². This suggests that a 10 fold higher concentration was accepted. We do note, however, that there may be future limitations to the use of pyocyanin. We are currently working on alternative redox mediators and stress regulons.

8. PAM is not explained in the manuscript. Defining the importance of the PAM site within the text is probably needed for the general audience of Nature. Likewise an explanation for adding the ribozyme sequences could be helpful.

We thank the reviewer for the comment. We have addressed this issue.

9. The strain and plasmid names are hard to follow in some figures. The worst case is probably Fig. 3D (NB101+pNB01+dCas-w vs. NB101+ptt01+pNB02+dCas9-w, etc.). Is there a simpler way to understand these results than listing each plasmid and host? Perhaps delta-soxS, no repression of soxS, S1 production, soxS regulated S1 production or something similar. I understand wanting to be exact about what was measured, but perhaps these details in the text or in the figure caption instead of the legend will make the paper easier to read.

We thank the reviewer for the comment. We have addressed this issue.

10. The first line of the abstract mentions "rapid and programmable information

transfer between electronics and biology”, but the responses shown appear to take hours and are likely of similar speed to adding a chemical inducer. The rate of information transfer and response is a very relevant topic here, so perhaps some attention could be given to this issue in the discussion. How could information exchange rates be increased?

The reviewer raises an important issue that has been raised by many in the information processing community. Biological information processing is much slower than in electronics and it is important to embrace this difference, taking advantage of the disparate time constants instead of shying away from them. We suggest that rapid and programmable information transfer in biological systems using redox is fast relative to studies involving DNA and evolutionary processes. Similarly, electronic to biological information processing via ion flow has been demonstrated for decades and these processes are deemed rapid and programmable. This is the case in neuromuscular electrical stimulation for example. In the case of deep brain stimulation, when electrical impulses are applied, there is rapid synapse formation in neuron junctions which results in release of neurotransmitters and the generation of Ca^{2+} waves in nearby astrocytes. These result in generation of gliotransmitters in the vicinity resulting in dilation of arteries and increased blood flow¹⁵. Our point being that when electrical signals are applied, there is a relay of information at all levels including metabolic, transcriptional, and biophysical that spans different cell types. The molecular information exchange here, which employs redox reactions at the electrode and subsequent molecular signalling events that occur via diffusion is naturally slower, but relative to evolutionary processes is quite fast. Most importantly, the means by which one can even transmit electronic information is now enabled.

11. The use of “etc.” several times on page 3 was distracting. If you state such as and list several things etc. does not seem needed.

We thank the reviewer for the comment. We have addressed this issue.

12. Designing genetic constructs for use in multiple chassis is a good goal for synthetic biology, and it does seem that modular tools such CRISPRa and CRISPRi will contribute to such goals. Showing that the construct works in Salmonella is worthwhile to report, but it should be noted in the discussion that *E. coli* and Salmonella are closely related species of bacteria. Although Salmonella is a “non-chassis strain”, it is not too surprising constructs in *E. coli* would be functional in Salmonella. The results do not suggest the ability to easily move these constructs into other species of bacteria.

The reviewer’s concern is noted and we chose *Salmonella* as it is a pathogenic strain as well as a non-chassis strain, yet it is genetically close to *E. coli*. *Salmonella* and *E. coli* have similar SoxS based stress defense response systems and our hypothesis was that in these strains repression of SoxS would result in increased protein expression from an electrically controlled SoxS promoter. To display this as well as to simplify our experiments, we used the same *E. coli* SoxS expressing gRNA’s (with just the changed in the first 15 bp specific to Salmonella SoxS), the same dCas9 promoters, same plasmids etc. Our point being that if a similar SoxS based stress defense response exists in a strain, depending on the availability of suitable plasmid origins and promoters, eCRISPR can be successfully

be transplanted to any strain of choice. We also suggest that those laboratories focused on expanding the CRISPR functions might find our approach appealing and additional species may be targeted.

Reviewer #3 (Remarks to the Author):

Summary:

This manuscript describes the expansion and improvement of a system previously developed by the Bentley lab for electrode-based control of gene expression.¹ The original system used a redox-sensitive SoxS promoter (regulated by the SoxR repressor) to control expression of a fluorescent protein via charge transfer from an electrode. Charge transfer is mediated by two small molecules, ferricyanide (Fcn) and pyocyanin (PYO). The major changes made here are to 1) add an additional layer of control by using the SoxS promoter to control expression of a CRISPR-based transcriptional activation, 2) enhance the dynamic range of the eCRISPR system by repressing the native SoxS response, and 3) transfer the eCRISPR system to *Salmonella enterica*. Overall, the eCRISPR system worked to induce expression in *E. coli* and *S. enterica*, although in *S. enterica*, electrical induction worked better without the added layer of CRISPRa control. This work represents an important next step in the development of electrode-controlled induction systems, but the manuscript would benefit from additional explanation of the overall engineering strategy, high level interpretation of results, and a brief, but clear outline of the electrode induction method in the results.

We thank the reviewer for the comments and we have incorporated the reviewer's comments and suggestions.

Major comments:

1. Overall, I found the manuscript somewhat difficult to follow. Addition of some explanatory statements throughout, and more descriptive plasmid and strain names would be helpful. A few examples:

We thank the reviewer for the comments and we have attempted to address the concerns.

a. It would be helpful at line 345 to state that the location of the gRNA binding site is what dictates whether dCas9- ω acts as an activator or repressor (i.e., binding in the promoter leads to CRISPRa, while binding after the TSS leads to CRISPRi).

We thank the reviewer for the suggestion. We have clarified.

b. Plasmid pTT01 could be referred to as pSoxS-phiLOV, similar to the pMC-phiLOV. Overall, replacing alphanumeric codes with more meaningful abbreviations would be useful throughout the manuscript.

We thank the reviewer for the suggestion. We have simplified all names to make it more understandable.

c. Lines 184-186: A more detailed explanation of what is the seed region of the

spacer and why point mutations would be expected to tune CRISPRi would be helpful.

We thank the reviewer for the suggestion. We have provided clarity in the manuscript and in Supplementary material.

d. Lines 293-295: At this point it would be helpful to clarify that repression of SoxS on the genome should not directly influence regulation of the SoxS promoters in the engineered constructs, because SoxR expression is unaffected.

We thank the reviewer for the suggestion. An earlier reviewer pointed to other problems that were clarified by noting this important point.

2. Why are both FCN and PYO necessary for activation? PYO can be re-oxidized abiotically at electrodes, so it is not clear why the additional mediator is needed. This may have been explained in the previous paper, but it should be briefly recapped here.

We thank the reviewer for the comments and we have now clarified.

3. Lines 148-149 indicate that cells were transferred back to aerobic conditions after charge transfer. Why was this done, and how long were cells incubated? If incubation times in figures refer to this post-charge transfer incubation, then the amount of time it took to do the charge transfer should also be noted. Overall, I think a clear description and diagram in the results section of the overall workflow for charge transfer experiments would be helpful.

We thank the reviewer for the comments. Upon electrical activation inside the anaerobic chamber that is maintained at room temperature, the cells were moved into the 37°C incubator that was also placed inside the anaerobic chamber. The only times when the cells were taken out of the anaerobic chamber after electrical activation was 1) during sampling for FACS and 2) obtaining conditioned media for AI-1 experiments.

With respect to the incubation times, the reviewer is correct that when a constant voltage is applied, the amount of time taken to discharge varying amounts of charge would be different in each case. However, these differences are quite small, for example the time taken to discharge 0.5C is ~3-4 minutes whereas for a charge of 0.1C, it is in the order of seconds. After the charges have been applied to cells that are maintained at room temperature inside the anaerobic chamber, when the cells are moved inside the 37°C incubator in the anaerobic chamber is only when the incubation times are started. So, the incubation times accurately reflect the time spent by the cells in 37°C incubator after the electrical activation. We have made changes in the methods sections to clarify these aspects.

4. Lines 273-279: If the 2-hr incubation and small amounts of charge transfer did not lead to strong induction of AI-1 production, why were these conditions chosen? A better explanation is needed for why experiments with the AI-1 output were done differently than experiments with the Philov output.

We have revised this section significantly, stressing the difference between low and high charge and the importance relative to information transfer.

5. The result in Figure 2E seems marginal. Significance testing may be appropriate here, it would be good to consult with a statistician. Also, please discuss potential reasons for the high leaky expression in the AI-1 system compared with the Philov system. It seems like the 0 charge transfer condition is well above the no AI-1 background.

As noted above, we have attempted to clarify this issue.

6. Lines 367-368: The analogy to a filter and amplifier are not clearly explained. Please state more clearly what is the input and what is the output in each analogy. It would also be helpful to explain how the filter and amplifier functions developed here could be integrated into larger engineered circuits.

We thank the reviewer for the comment. We have clarified and restructured the paper on the topics of information processing aspects including a new fig. 4 and why they are important.

7. Lines 458-467: Application of this system to gut microbiomes seems very far in the future. Are there any medium-term applications that could also be discussed here? Nearer-term application of eCRISPR in a biotech setting seems more feasible.

We have provided additional context while retaining the microbiome theme. Specifically, utility in gut-on-a-chip issues and the design of probiotic synbio constructs are now included.

8. The methods do not provide sufficient information about how charge transfer was controlled. If a constant voltage was applied for a pre-determined amount of time, how were different charge transfer amounts achieved? This relates back to point 3.

We have clarified this issue above.

9. For figure S10, why was a combination of Fcn and PYO used, instead of just PYO? Again, this is where an explanation of the specific roles of Fcn and PYO in the system would be helpful.

We have clarified this issue above as well as in the manuscript.

Minor comments:

1. All potentials noted in the manuscript should indicate what they are relative to, i.e., is it 0.5 VSHE or 0.5 VAg/AgCl?

We thank the reviewer for the comment. All voltages are relative to Ag/AgCl and we have amended the same in the methods section.

2. There are a few places in the manuscript where symbols were replaced by blocks: e.g., Line 270, Line 397.

We have corrected the mistake.

3. There are inconsistencies throughout the manuscript whether there is a space between a number and a unit.

We have addressed this issue.

4. Line 339: The text indicates that DJ901 is a Δ soxRS strain, but Figure 3 indicates that it is a Δ soxS strain. Please correct.

We have addressed this issue.

5. Line 414: change 'be' to 'by'
We thank the reviewer for the comment.

6. Line 434: remove 'While'
We thank the reviewer for the comment.

7. Line 444-445: I recommend softening the language here. I'm not sure that use of two redox mediators should be considered 'direct connection between electronic signals and bacterial cells'.

The reviewer raises a point similar to point 10 of reviewer 2. We have improved our manuscript which clarifies and also highlights distinct differences between our electrical activation and that of neuromuscular electrical stimulation, for example. While our society has benefited for decades by the connection between an applied voltage and biological activity, the connection has been limited to ionic currents. There has been almost no direct connection between electronics and the *molecular* information that is pervasive in biology. One noted counter example is the glucose analyser used for treating diabetes. Interestingly this is redox mediated. While we have become tremendously sophisticated with respect to ionic information transfer, we have made almost no advances exploiting communication between electronics and the *molecules* of biology. We suggest here and in many of our other papers that redox is a medium that embraces and accommodates both electron movement and the structural information that is embedded in molecules. In our opinion, tapping into redox will enable significant advances in metabolic engineering and perhaps usher in a completely novel means for the detection of disease and even treatment. For example, we have recently shown that oxidative stress levels in blood serum, electrochemically detected by the purposeful addition of redox mediators, can be distinguished in samples from patients with schizophrenia relative to "healthy" controls^{16,17}. There is currently no such data used in the diagnosis and treatment of schizophrenia, nor to the best of our knowledge, any other mental disorder. Hence, we feel redox-based detection and control methodologies, especially those that build on (1) advanced genetic techniques such as CRISPR and (2) that append these to electronics and signal processing are sorely needed and potentially revolutionary.

Certainly, the above description is grand, far too grand to include in our manuscript, but we hope this conveys our conviction to these methodologies and

excitement about this work. That we include it here, in our Comments to Reviewers' Concerns section, remains relevant as these rebuttal comments are included in a publication in *Nature Communications*. This is one of many motivating factors for targeting this journal.

References:

- 1 Tsao, C. Y., Hooshangi, S., Wu, H. C., Valdes, J. J. & Bentley, W. E. Autonomous induction of recombinant proteins by minimally rewiring native quorum sensing regulon of *E. coli*. *Metab Eng* **12**, 291-297, doi:10.1016/j.ymben.2010.01.002 (2010).
- 2 Tschirhart, T. *et al.* Electronic control of gene expression and cell behaviour in *Escherichia coli* through redox signalling. *Nat Commun* **8**, 14030, doi:10.1038/ncomms14030 (2017).
- 3 Dong, C., Fontana, J., Patel, A., Carothers, J. M. & Zalatan, J. G. Synthetic CRISPR-Cas gene activators for transcriptional reprogramming in bacteria. *Nature Communications* **9**, 2489, doi:10.1038/s41467-018-04901-6 (2018).
- 4 Li, Z. & Demple, B. SoxS, an activator of superoxide stress genes in *Escherichia coli*. Purification and interaction with DNA. *Journal of Biological Chemistry* **269**, 18371-18377 (1994).
- 5 Varghese, S., Tang, Y. & Imlay, J. A. Contrasting Sensitivities of *Escherichia coli* Aconitases A and B to Oxidation and Iron Depletion. *Journal of Bacteriology* **185**, 221-230, doi:10.1128/jb.185.1.221-230.2003 (2003).
- 6 Liochev, S. I. & Fridovich, I. Modulation of the Fumarases of *Escherichia coli* in Response to Oxidative Stress. *Archives of Biochemistry and Biophysics* **301**, 379-384, doi:<https://doi.org/10.1006/abbi.1993.1159> (1993).
- 7 Pomposiello, P. J. & Demple, B. in *Advances in Microbial Physiology* Vol. 46 319-341 (Academic Press, 2002).
- 8 Zheng, M., Doan, B., Schneider, T. D. & Storz, G. OxyR and SoxRS Regulation of *fur*. *Journal of Bacteriology* **181**, 4639-4643 (1999).
- 9 Gaudu, P., Moon, N. & Weiss, B. Regulation of the soxRS Oxidative Stress Regulon: REVERSIBLE OXIDATION OF THE Fe-S CENTERS OF SoxR IN VIVO. *Journal of Biological Chemistry* **272**, 5082-5086, doi:10.1074/jbc.272.8.5082 (1997).
- 10 Zheng, M., Åslund, F. & Storz, G. Activation of the OxyR Transcription Factor by Reversible Disulfide Bond Formation. *Science* **279**, 1718-1722, doi:10.1126/science.279.5357.1718 (1998).
- 11 Sadat Mousavi, P. *et al.* A multiplexed, electrochemical interface for gene-circuit-based sensors. *Nat Chem* **12**, 48-55, doi:10.1038/s41557-019-0366-y (2020).
- 12 Hirose, A., Kouzuma, A. & Watanabe, K. Towards development of electrogenetics using electrochemically active bacteria. *Biotechnol Adv* **37**, 107351, doi:10.1016/j.biotechadv.2019.02.007 (2019).
- 13 Srivastava, R., Cha, H. J., Peterson, M. S. & Bentley, W. E. Antisense downregulation of sigma(32) as a transient metabolic controller in *Escherichia coli*: effects on yield of active organophosphorus hydrolase. *Appl Environ Microbiol* **66**, 4366-4371, doi:10.1128/aem.66.10.4366-4371.2000 (2000).
- 14 Srivastava, R., Peterson, M. S. & Bentley, W. E. Stochastic kinetic analysis of the *Escherichia coli* stress circuit using sigma(32)-targeted antisense. *Biotechnol Bioeng* **75**, 120-129, doi:10.1002/bit.1171 (2001).

- 15 McIntyre, C. C. & Anderson, R. W. Deep brain stimulation mechanisms: the control of network activity via neurochemistry modulation. *J Neurochem* **139 Suppl 1**, 338-345, doi:10.1111/jnc.13649 (2016).
- 16 Kim, E. *et al.* Validation of oxidative stress assay for schizophrenia. *Schizophr Res* **212**, 126-133, doi:10.1016/j.schres.2019.07.057 (2019).
- 17 Kang, M. *et al.* Reliable clinical serum analysis with reusable electrochemical sensor: Toward point-of-care measurement of the antipsychotic medication clozapine. *Biosensors and Bioelectronics* **95**, 55-59, doi:<https://doi.org/10.1016/j.bios.2017.04.008> (2017).

Reviewers' Comments:

Reviewer #1:

Remarks to the Author:

The authors have satisfied all of this reviewer's and his/her fellow reviewers' concerns.

Reviewer #2:

Remarks to the Author:

The authors have made several improvements in the revised manuscript. The response to reviewers was thoughtfully prepared and the authors addressed most of the reviewer's concerns. The new experimental measurements presented in Figure 4 were clever and demonstrate the utility of the reported constructs in many applications.

I may be in the minority view here, but in the end, despite the strong revision, the paper is still an incremental advance in the field. Lines 547-556 give a great summary of the work that was done as part of this study. The overall impact of the results still comes up short as compared to more innovative/breakthrough findings that are typical of a publication in Nature Communications. The work was carefully done and interpreted correctly, but I fail to see this work as more than a solid follow-up to the author's previous work. I appreciate the non-trivial work was done to troubleshoot the integration of existing genetic circuitry with the previously reported electrogenetic promoters, but the progress remains incremental even with the experimental work added to the revision. My second major concern (as discussed below) is that the revised manuscript still tries to emphasize rather tenuous connections between the findings and concepts of information exchange in a way that is non-rigorous and potentially misleading. For these reasons the manuscript is more appropriate for a specialized journal in synthetic biology

Major comment:

1. The incorporation of terms related to information theory and communication networks remains underdeveloped and a weakness of the manuscript. These concepts are still used as buzz words that do not improve the reader's understanding of the findings. Here are several examples:

a. Genomic noise is poorly defined. Is this leaky expression, unknown feedback mechanisms within the cell that result in an unexpected change in the expression of a particular gene? There isn't any analysis of noise in the manuscript, so the meaning of this term (which I have not encountered previously) should be clarified.

b. The axis of Scheme 1 is "molecular information transfer". How is this quantified? The amount of information transferred is a linear function of signal concentration? These concepts of information flow seem too non-specific. Why not label this axis "Signal concentration" or "Input signal detected by cell".

c. Line 53: "Moreover, this modality conforms nicely with the theoretical underpinnings of information transfer."

Certainly any cellular network that senses and responds to an input signal could be analyzed using concepts from information theory. Putting this type of claim in the first paragraph implies a unique connection between electrically controlled gene expression and theoretical concepts of information theory, but the same underpinnings would apply to gene networks that respond to any external input (chemical, pressure, temperature).

d. Line 76 "In this work, we have expanded the range of and extent (ie., "bandwidth") by which electronic information can be transmitted to biological systems by creating a redox-mediated

device-bio communication channel based on the CRISPR-mediated tool set that enables multiplexed host genome control.”

What is meant by range and extent of information transfer in this context? Range is the spatial range? The range of concentrations detected by the cell? If bandwidth (which appears to have few definitions) could be defined as the rate of information transfer from an electronic source to the biological system, I suppose the extent of information is increased compared to a cell that does not have the reported genetic constructs. But, isn't that the same as taking a circuit that responds to aTc and adding a new module that responds to IPTG and claiming an increase in bandwidth? Unless information transfer is being quantified in some way (channel capacity, bits of information being exchanged per time) it is not appropriate to discuss the bandwidth of the reported system.

e. Line 141: “Cells with eCRISPR circuits displayed enhanced signal strength as well as reduced “noise” relative to cells without eCRISPR control.”

I don't see reasonable discussion or quantitative analysis of noise in this manuscript. This claim is not well supported by the analysis presented.

Minor Comments:

1. On line 535, was a 0.3 V or a 0.5 V field applied? Figure 4D mentions a 0.5 V field.

2. Figure 4C, there is a box instead of an omega in the legend (dCas9-box).

3. In Figure S2, the figure caption refers to “spacers (105, 108, and 66 spacer)” and the figure shows control spacer, 105 spacer, and 108 spacer. I would conclude the control spacer is the one of length 66 bp, but that would be my guess.

4. Line 107: “we adapted the previously described bacterial CRISPRa system wherein a dCas9- ω fusion was used as a transcriptional activator resulting in 23-fold transcriptional activation.”

The authors should clarify that the 23-fold transcriptional activation was observed in the previous study. The wording here is slightly misleading.

5. Line 117: “In addition to building this E. coli-specific electrogenetic controller, we sought to demonstrate function across a variety of non-chassis strains”

Again, a little misleading. A variety of non-chassis strains sounds like multiple strains, whereas in reality the controller was proven to work in a single non-chassis strain.

6. The manuscript remains slightly difficult to follow. This is partly due to fact that many different constructs are used with minor (yet important) differences. The revised article was much improved in this respect, but reading the manuscript still requires constant checking and rechecking of the differences in all the guide RNAs, Cas proteins, etc.

7. I am not convinced by the response “We suggest that rapid and programmable information transfer in biological systems using redox is fast relative to studies involving DNA and evolutionary processes.” So the gene circuit is fast relative to evolutionary timescales? Evolutionary time scales refers thousands to millions of years or perhaps the generation time? This seems like a weak argument. If the input information is funneled into a transcription factor networks, there is no speed or programmability advantages of the redox responsive system reported relative to chemical induction.

Reviewer #3:

Remarks to the Author:

My comments have been sufficiently addressed.

Reviewer #2 (Remarks to the Author):

The authors have made several improvements in the revised manuscript. The response to reviewers was thoughtfully prepared and the authors addressed most of the reviewer's concerns. The new experimental measurements presented in Figure 4 were clever and demonstrate the utility of the reported constructs in many applications.

I may be in the minority view here, but in the end, despite the strong revision, the paper is still an incremental advance in the field. Lines 547-556 give a great summary of the work that was done as part of this study. The overall impact of the results still comes up short as compared to more innovative/breakthrough findings that are typical of a publication in Nature Communications. The work was carefully done and interpreted correctly, but I fail to see this work as more than a solid follow-up to the author's previous work. I appreciate the non-trivial work was done to troubleshoot the integration of existing genetic circuitry with the previously reported electrogenetic promoters, but the progress remains incremental even with the experimental work added to the revision. My second major concern (as discussed below) is that the revised manuscript still tries to emphasize rather tenuous connections between the findings and concepts of information exchange in a way that is non-rigorous and potentially misleading. For these reasons the manuscript is more appropriate for a specialized journal in synthetic biology

We thank the reviewer for taking the time to thoughtfully prepare his/her critiques below. In general, the broad concept conveyed in this work is that we are removing key barriers that limit the use of redox signaling to bridge electronics and biology. Importantly, this concept has not been demonstrated previously and is a result of expanding the well-known utility of CRISPR to include multiplexed functions that reduce pleiotropic noise, in particular, noise that stems from the use of redox signaling. As a result, our work offers a new type of engineering control that is broadly applicable to all cells within molecular communication range of an electrode. These changes were not trivial and open a wide-array of new applications, the last of which we demonstrate as a novel, programmable method to create gradients of gene expression in a single device. Second, we have clarified the connections between communication theory and biology and offer that there are precious few experimental studies that offer tractable methodologies upon which information theorists can test hypotheses and provide alternative biological trajectories based on information theory. Finally, we address each of the comments below and have made corresponding alterations to the manuscript.

Major comment:

1. The incorporation of terms related to information theory and communication networks remains underdeveloped and a weakness of the manuscript. These concepts are still used as buzz words that do not improve the reader's understanding of the findings. Here are several examples:

a. Genomic noise is poorly defined. Is this leaky expression, unknown feedback mechanisms within the cell that result in an unexpected change in the expression of a particular gene? There isn't any analysis of noise in the manuscript, so the meaning of this term (which I have not encountered previously) should be clarified.

We thank the reviewer for this comment. To address this issue, we will attempt to clarify the central question “what is noise?” and how is noise defined in context of cellular networks and information theory. Most importantly, we have provided modifications in the manuscript that make these points clear.

The concept of noise in biologic networks is not new^{1, 2}. Noise can be defined as the random stochasticity that exists in the biological networks and this randomness can manifest itself in many ways including cell to cell variability in transcription rates, translation rates, etc. Stochasticity can be attributed to internal, as well as external noise in the system. Biological systems can eliminate this noise and function with precise regularity, and this is achieved using positive and negative feedback loops in gene circuits that serve to reduce the randomness. In the context of electro-responsive promoter systems, these systems are primarily derived from oxidative stress response promoters. Induction of these promoters mediates oxidative stress in cells and to counter this, cells have their own feedback mechanisms (i.e. SoxS derived oxidative stress responses). While it is a positive feedback from the cell’s perspective, from our perspective, we see these unintentional, pleiotropic alterations as a negative feedback that invariably leads to loss in signal intensity.

From an information processing perspective, noise can be defined as any factor that impedes the flow of information from sender to receiver. In our system, the repression of the oxidative stress response (suppression of noise) leads to increased signal intensity. Moreover, the actual oxidative stress defense response, or noise, is a mixture of metabolic and many ‘omics interactions, all arising from the activated transcription factor, SoxS. Hence, we refer to this noise as the genomic noise. We must add that, pinpointing each individual component of what caused the noise has not been a focus of our work – as the SoxS-derived oxidative stress response has been well-studied. Instead, we demonstrate that by using CRISPR to prevent the oxidative stress response, in all its facets, effectively a stronger correlation between input and output can be achieved. More importantly, because CRISPR can multiplex and simultaneously amplify and inhibit gene expression, our work for the first time shows how CRISPR can reduce noise while ensuring information transfer.

b. The axis of Scheme 1 is “molecular information transfer”. How is this quantified? The amount of information transferred is a linear function of signal concentration? These concepts of information flow seem too non-specific. Why not label this axis “Signal concentration” or “Input signal detected by cell”.

To make this simple, we have altered Figure 1 to “Redox Signal”.

c. Line 53: “Moreover, this modality conforms nicely with the theoretical underpinnings of information transfer.”

Certainly any cellular network that senses and responds to an input signal could be analyzed using concepts from information theory. Putting this type of claim in the first paragraph implies a unique connection between electrically controlled gene expression and theoretical concepts of information theory, but the same underpinnings would apply to gene networks that respond to any external input (chemical, pressure, temperature).

The reviewer is correct in pointing out that any network of communication can be analyzed with information theory and pH, temperature osmosis, heat shock and many other factors could be explained in the same way. These interpretations are rare. To the best of our knowledge, information theory has

not been used in the context of synthetic biology, where from the bottom up, circuits are constructed based on principles of information transmission. Moreover, in our work, direct information transfer from electronics to biology is demonstrated and this revolves around the recognition that redox can mediate this information exchange.

Redox communication has a unique connection in that its major components each resemble the transmitter-channel-receiver model. Simply adding chemical components to cell media is, according to theory, a form of communication, but we are using programmable electrical signals to enable chemical transmissions away from an electrode and we have constructed cells that uniquely receive this transmitted information. Re-creating this type of communication channel, without direct electronic input, would require substantial microfluidic designs to spatiotemporally control signal gradients. This would not be broadly applicable. Moreover, the electrochemical signals that we generate are both expendable as well as recyclable. That is, the oxidized Fcn that is generated at the electrode acts as a signal carrier to the cell that later gets reduced by the cell. However, we can pass another set of information once again by another round of oxidation using the now reduced Fcn. By virtue of this recyclability, we can build ON/OFF switches, stimulate oscillatory behaviors in biological systems, etc., such controlled inputs cannot be achieved using chemical inducers.

d. Line 76 “In this work, we have expanded the range of and extent (ie., “bandwidth”) by which electronic information can be transmitted to biological systems by creating a redox-mediated device-bio communication channel based on the CRISPR-mediated tool set that enables multiplexed host genome control.”

What is meant by range and extent of information transfer in this context? Range is the spatial range? The range of concentrations detected by the cell? If bandwidth (which appears to have few definitions) could be defined as the rate of information transfer from an electronic source to the biological system, I suppose the extent of information is increased compared to a cell that does not have the reported genetic constructs. But, isn't that the same as taking a circuit that responds to aTc and adding a new module that responds to IPTG and claiming an increase in bandwidth? Unless information transfer is being quantified in some way (channel capacity, bits of information being exchanged per time) it is not appropriate to discuss the bandwidth of the reported system.

We accept that this term has many definitions and to reduce confusion we have removed it from the manuscript. In addition, we have clarified the referenced section (now lines 94-96) of the manuscript to illustrate that what we have removed key barriers to communication between electronic devices and biology. These barriers go beyond simply adding new modules to cells – these barriers included the broad applicability of CRISPR to electrogenetic control, the use of CRISPR to allow minimally-engineered hosts to participate in communication networks, and the ability of cells to sense and respond to electrically transmitted chemical signals.

e. Line 141: “Cells with eCRISPR circuits displayed enhanced signal strength as well as reduced “noise” relative to cells without eCRISPR control.”

I don't see reasonable discussion or quantitative analysis of noise in this manuscript. This claim is not well supported by the analysis presented.

We have addressed this criticism above under the reviewer's comment 1a. We have also correspondingly altered the paragraph from line 153-172 to explain our position.

Minor Comments:

1. On line 535, was a 0.3 V or a 0.5 V field applied? Figure 4D mentions a 0.5 V field.

We thank the reviewer for finding this error. We have corrected Figure 4D to read 0.3 V.

2. Figure 4C, there is a box instead of an omega in the legend (dCas9-box).

Thank you for this correction, we have made the corresponding change.

3. In Figure S2, the figure caption refers to "spacers (105, 108, and 66 spacer)" and the figure shows control spacer, 105 spacer, and 108 spacer. I would conclude the control spacer is the one of length 66 bp, but that would be my guess.

Thank you for this correction, we have changed the name back to 66 spacer.

4. Line 107: "we adapted the previously described bacterial CRISPRa system wherein a dCas9- ω fusion was used as a transcriptional activator resulting in 23-fold transcriptional activation."

The authors should clarify that the 23-fold transcriptional activation was observed in the previous study. The wording here is slightly misleading.

We have altered the referenced sentence to emphasize

5. Line 117: "In addition to building this *E. coli*-specific electrogenetic controller, we sought to demonstrate function across a variety of non-chassis strains"

Again, a little misleading. A variety of non-chassis strains sounds like multiple strains, whereas in reality the controller was proven to work in a single non-chassis strain.

We have altered the wording to emphasize that we used CRISPR to modify both WT *E. coli* and *S. enterica serovar typhimurium*. Previously, only the *E. coli* strains harboring a *soxS* knockout (i.e. strain DJ901) could be electrogenetically controlled.

6. The manuscript remains slightly difficult to follow. This is partly due to fact that many different constructs are used with minor (yet important) differences. The revised article was much improved in this respect, but reading the manuscript still requires constant checking and rechecking of the differences in all the guide RNAs, Cas proteins, etc.

With regards to readability, we believe the reviewer is correct that there are many constructs with small, but important, differences. To the degree that it was possible, we have made sure to add clarifying descriptions of each plasmid when describing their use within an experiment.

7. I am not convinced by the response "We suggest that rapid and programmable information transfer in biological systems using redox is fast relative to studies involving DNA and evolutionary processes." So the gene circuit is fast relative to evolutionary timescales? Evolutionary time scales refers thousands to millions of years or perhaps the generation time? This seems like a weak argument. If the input

information is funneled into a transcription factor networks, there is no speed or programmability advantages of the redox responsive system reported relative to chemical induction.

Our previous comment about DNA and evolution was poorly stated. Our point relative to the speed by which information can be conveyed to biology was misinterpreted. We suggest that electronic transmission is far superior to the addition of inducers. In the cases of neuromuscular and neural stimulation, the information flow is rapid and expansive, as are the biological responses. With the opening and closing of ion channels, further downstream processes are complex with changes in metabolic profiles, proteomic and transcriptional changes often spanning across a multitude of cell systems before resulting in a desired output. In these kinds of systems with highly complex circuitry, it is very hard to build bottom-up circuits imbibing synthetic biology principles.

In this work, bottom-up assembly starts with a singular electro-responsive promoter and, based on the intended gene targets and the assembled genetic regulatory circuits, diversity is added as a part of the design. Here, we introduced the built-in functions of CRISPR. In this way, CRISPR adds a further layer of sophistication and control in information processing by reducing noise and amplifying signal, etc. Importantly, these functions are actuated in a spatiotemporally controlled way, using electronics.

References

1. Gardner, T.S. & Collins, J.J. Neutralizing noise in gene networks. *Nature* **405**, 520-521 (2000).
2. Hasty, J., Pradines, J., Dolnik, M. & Collins, J.J. Noise-based switches and amplifiers for gene expression. *Proc Natl Acad Sci U S A* **97**, 2075-2080 (2000).